# Autophagy is a gatekeeper of hepatic differentiation and carcinogenesis by controlling the degradation of Yap

Youngmin A. Lee[1,2], Luke A. Noon[1,3], Kemal M. Akat[2], Maria D. Ybanez[1], Ting-Fang Lee[1], Marie-Luise Berres[4,5], Naoto Fujiwara[1,6], Nicolas Goossens[1,7], Hsin-I Chou[1], Fatemeh P. Parvin-Nejad[1], Bilon Khambu[8], Elisabeth G.M. Kramer[9], Ronald Gordon[10], Cathie Pfleger[5], Doris Germain[11], Gareth R. John[9], Kirk N. Campbell[12], Zhenyu Yue[9], Xiao-Ming Yin[8], Ana Maria Cuervo [13], Mark J. Czaja[14], M. Isabel Fiel[10], Yujin Hoshida [1,6] & Scott L. Friedman [1]

Activation of the Hippo pathway effector Yap underlies many liver cancers, however no germline or somatic mutations have been identified. Autophagy maintains essential metabolic functions of the liver, and autophagy-deficient murine models develop benign adenomas and hepatomegaly, which have been attributed to activation of the p62/Sqstm1-Nrf2 axis. Here, we show that Yap is an autophagy substrate and mediator of tissue remodeling and hepatocarcinogenesis independent of the p62/Sqstm1-Nrf2 axis. Hepatocyte-specific deletion of Atg7 promotes liver size, fibrosis, progenitor cell expansion, and hepatocarcinogenesis, which is rescued by concurrent deletion of Yap. Our results shed new light on mechanisms of Yap degradation and the sequence of events that follow disruption of autophagy, which is impaired in chronic liver disease.

[1] Division of Liver Diseases, Icahn School of Medicine at Mount Sinai, New York, NY 10029, USA. [2] Laboratory of RNA Molecular Biology, Rockefeller University, New York, NY 10065, USA. [3] CIBERDEM, Centro de Investigación Príncipe Felipe, 46012 Valencia, Spain. [4] Department of Internal Medicine III, University Hospital RWTH Aachen, 52074 Aachen, Germany. [5] Department of Oncological Sciences, Icahn School of Medicine at Mount Sinai, New York, NY 10029, USA. [6] Division of Digestive and Liver Diseases, University of Texas Southwestern Medical Center, Dallas, Texas Tx 75390, USA. [7] Division of Gastroenterology and Hepatology, Geneva University Hospital, 1205 Geneva, Switzerland. [8] Department of Pathology & Laboratory Medicine, Indiana University School of Medicine, Indianapolis, IN 46202, USA. [9] Department of Neurology, Friedman Brain Institute, Icahn School of Medicine at Mount Sinai, New York, NY 10029, USA. [10] Department for Pathology, Icahn School of Medicine at Mount Sinai, New York, NY 10029, USA. [11] Department of Hematology and Medical Oncology, Icahn School of Medicine at Mount Sinai, New York, NY 10029, USA. [12] Division of Nephrology, Icahn School of Medicine at Mount Sinai, NY 10029 New York, USA. [13] Department of Developmental and Molecular Biology, Institute for Aging Studies, Albert Einstein College of Medicine, Bronx, NY 10461, USA. [14] Department of Medicine, Emory University School of Medicine, Atlanta, GA GA 30307, USA. Correspondence and requests for materials should be addressed to Y.A.L. (email: ylee04@rockefeller.edu)

Hepatocellular carcinoma (HCC) is the third leading cause of cancer-related mortality worldwide[1] with a rising incidence attributed to advanced liver disease from multiple etiologies[2]. HCC develops as a multistep process on a background of chronic liver injury leading to inflammation, stromal activation, fibrosis, and regeneration[3]. Pathways involved in the malignant transformation of hepatocytes from dysplastic nodules to early stage HCC include oxidative stress responses, deregulation of the protein folding machinery and dedifferentiation with reexpression of fetal genes[4,5].

Autophagy, a nexus of metabolic homeostasis in liver is typically impaired during chronic injury, leading to reduced clearance of cellular constituents, and dysregulated mitochondrial and cellular integrity[6–8]. Although reduced autophagy is a feature of chronic liver disease, mechanisms linking autophagy loss to carcinogenesis are not fully clarified[1–5,8–13].

Autophagy has a dual role in cancer. During tumor initiation autophagy removes damaged organelles and reactive oxygen species to maintain genomic stability and promote oncogene-induced senescence, thereby inhibiting malignant transformation. However, in advanced tumors and metastases autophagy may promote tumor cell survival in low-nutrient conditions and chemotherapy-induced stress[14,15].

Autophagy's role in liver is primarily tumor-suppressive, yet its full activity in hepatocarcinogenesis is unclear. Rodent models with liver-specific deletion of the autophagy-related proteins 5 (Atg5)[16] or Atg7[17–19] display hepatic metabolic dysfunction with steatosis, ER stress and marked hepatomegaly and develop benign tumors (adenomas), but not HCCs. Central to this phenotype is signaling by p62/Sqstm1-Keap1-Nrf2, because loss of either p62/Sqstm1[20] or Nrf2 in autophagy-deficient double knock out mice normalizes liver size[16,21] and decreases tumorigenesis in p62/Sqstm1/Atg7 DKO mice[18]. When autophagy is impaired, the autophagy adaptor protein p62/Sqstm1 accumulates and binds Kelch-like protein 1 (KEAP1), a negative regulator of the oxidative master regulator nuclear factor erythroid 2-related factor 2 (NRF2, NFE2L2). NRF2 stabilization leads to a deleteriously high antioxidative response with induction of downstream targets such as NAD(P)H quinone oxireductase 1 (Nqo1) or gluthathion S-transferase 1 (Gst1) exacerbating liver injury and metabolic reprogramming[22]. The p62/Sqstm1-Keap1-Nrf2 axis has been thus identified as the pathway linking autophagy inhibition in the liver to hepatomegaly and tumorigenesis. However, the current paradigm does not account for the dramatic tissue remodeling and progenitor cell activation within autophagy deficient livers prior to the onset of cancer.

Yap is the major nuclear effector of the Hippo signaling pathway and mediates organ size control, proliferation, differentiation, and stemness. Yap can be activated by multiple stimuli including DNA damage[23], ER stress[24], and mechanical/shear stress[25], but its inhibition is tightly regulated and includes signaling by the Hippo core kinase cassette[26]. Yap may also be inhibited by cytoplasmic sequestration by dystrophin–glycoprotein complex[27] or Yap/Amot, which can result in the translocation of Yap to the cytoplasm or to cell junctions[28,29]. Upon its activation, Yap translocates into the nucleus, where it interacts with Tead1–4 as transcriptional co-activators to promote a range of biologic activities, including cell survival, proliferation, polarity, and, most importantly, organ size control[30]. Liver-specific deletions of Hippo pathway components (e.g., Mst1/2, Sav1, Nf2) or overexpression of Yap, lead to prominent hepatomegaly, progenitor cell expansion and hepatocarcinogenesis[31–34]. Ectopic overexpression of Yap in hepatocytes promotes their plasticity, with dedifferentiation to a cholangiocytic phenotype[35].

Yap activation is an early event in liver cancer development[36]. Up to 65% of HCCs harbor dysregulation of the Hippo/Yap pathway, which is associated with a significantly poorer prognosis[37,38]. Strikingly, despite comprehensive efforts, no germline or somatic mutations have been uncovered[39]. Thus, mechanisms underlying Yap dysregulation in hepatocarcinogenesis remain obscure.

Here, we identify Yap as an autophagy substrate and as an essential downstream mediator of tissue remodeling, progenitor cell activation and hepatocarcinogenesis in autophagy-deficient livers. Our results shed new light on mechanisms of Yap dysregulation in HCC by implicating impaired autophagy as a potential driver of Yap stabilization and activity. Because autophagy is impaired chronic liver disease, the findings provide a potential basis for enhanced Yap activity associated with hepatocarcinogenesis[7,40].

## Results

**Autophagy maintains hepatic organ size and differentiation.** To explore the mechanisms underlying autophagy's regulation of hepatic growth and tumorigenesis, we generated mice with conditional deletion of Atg7 in hepatocytes (Albumin-CRE: Atg7[F/F] mice) (Fig. 1a). Consistent with earlier studies[18,19,41] mice with liver-specific deletion of the autophagy-related protein 7 (Atg7) displayed massive hepatomegaly with up to 8.5 fold increased relative liver weight compared to CRE negative littermates at 3 months of age (Supplementary Fig. 1A, B, C). By histological analysis, hypertrophic and hyperplastic hepatocytes were prominent with a marked thickening of hepatocyte plates that were composed of at least 2–3 cells instead of single cells based on HE staining and on the pattern of cell membranous staining for β-catenin (Supplementary Fig. 1D, F). Hepatocytes were markedly dedifferentiated based on HNF4α staining (Fig. 1b) and whole liver RNA array analysis in which hepatocyte-specific genes including Albumin, Transthyretin and complement factors (C2, C3) were significantly downregulated (Fig. 1c). Immunostaining showed Epcam+ progenitor cell expansion (Fig. 1d). Blinded histologic scoring demonstrated increased lobular and portal inflammation, steatosis and ballooning (Supplementary Fig. 1E) and significant fibrosis (Supplementary Fig. 1G, H). Hepatocyte proliferative activity was markedly increased as assessed by Ki67 staining and quantification (Supplementary Fig. 1I, J). All Atg7 KO mice developed dysplastic nodules at 8 months, which progressed to HCC at 12 months (Supplementary Fig. 2A, B, C). In contrast to previous reports[17], tumors had typical features of HCCs including pseudoglandular transformation, loss of reticulin staining and diffuse Gst1 staining[42] (Supplementary Fig. 2D, E).

**Autophagy maintains hepatic Hippo tumor suppressor pathway.** Yap is a key regulator of hepatocellular fate[35], stem cell renewal, and liver size with overexpression or dysregulation of the Hippo/Yap signaling pathway leading to liver outgrowth, progenitor cell expansion and tumorigenesis[32,34]. We thus analyzed livers of Atg7 KO mice at 3 months and tumors at 12 months of age and found increased cytoplasmic and nuclear Yap (Fig. 1e). consistent with activation of the Hippo pathway effector. Primary hepatocytes isolated from Atg7 KO mice also displayed increased nuclear Yap staining (Supplementary Fig. 3B). Gene set enrichment analysis (GSEA) of the livers' transcriptome revealed significant enrichment of Yap activation signatures previously identified in Mst1/Mst2 KO mice[43] and in human cancer cells[44,45] (Fig. 1f). Significant induction of genes associated with Yap activation and Yap target genes was confirmed by gene array (Fig. 1g) and qRT-PCR (Fig. 1h).

To analyze the immediate response to Atg7 deletion, we generated an inducible conditional Atg7 KO mice by crossing

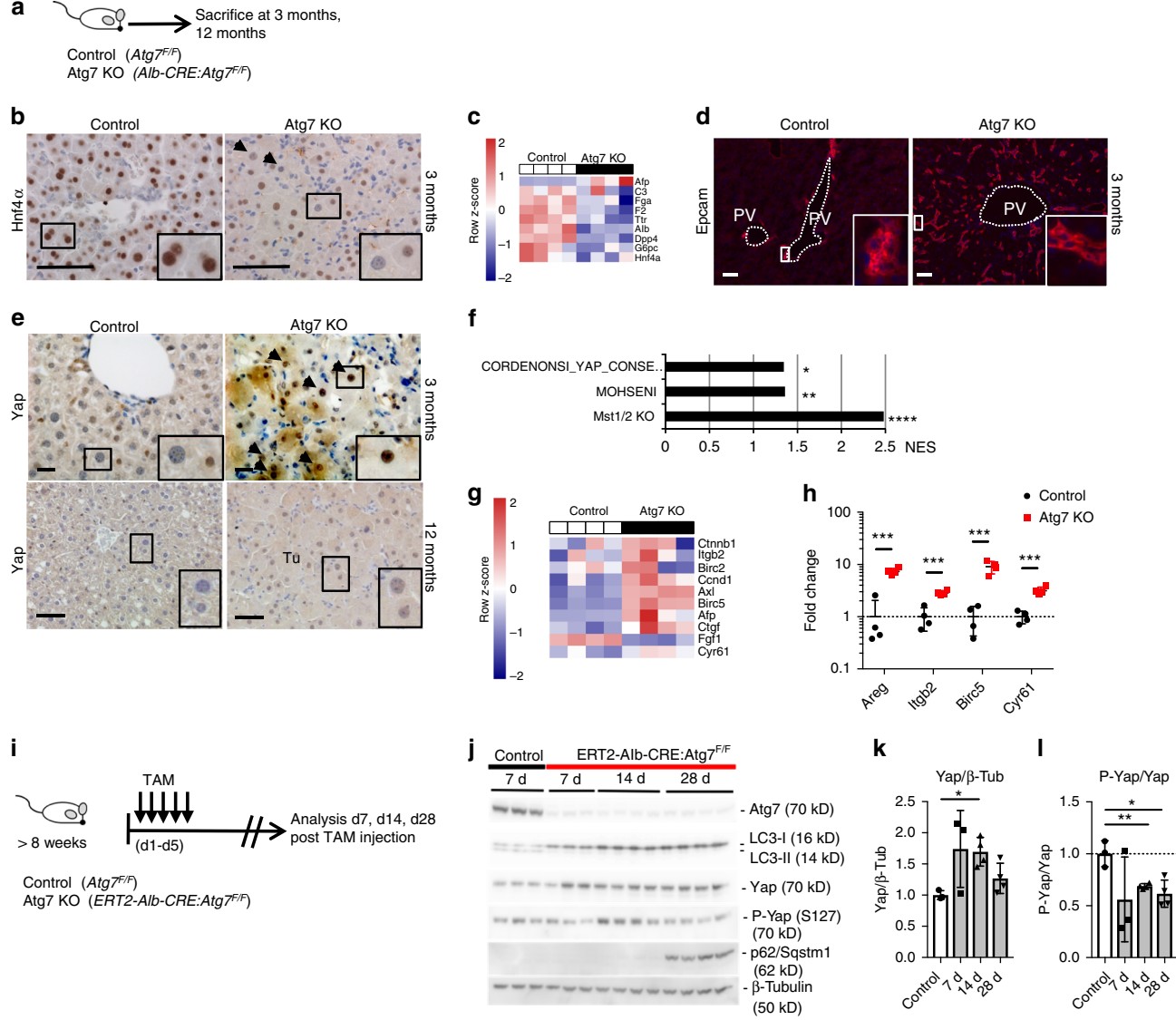

**Fig. 1** Autophagy maintains Hippo/Yap pathway in liver. **a** Control (*Atg7^F/F*) and Atg7 KO mice (*Alb-CRE:Atg7^F/F*) were analyzed at 3 months ($n = 4$) and 12 months of age ($n = 6$ and 7 respectively). **b** Immunostaining for HNF4α in control and Atg7 KO mice. **c** Gene expression array analysis of whole liver RNA from controls and Atg7 KO animals. Data indicate row z-scores. **d** Immunostaining for Epcam in control and Atg7 KO mice. **e** Immunostaining for Yap in control and Atg7 KO liver sections at 3 months and 12 months. Tu, tumor. **f** Gene set enrichment analysis for enrichment of Hippo/Yap gene signatures and plotting of NES scores. NOM *P*-value *$P < 0.05$, **$P = 0.001$, ****$P < 0.0001$. NES, normalized enrichment score. **g** Gene expression array analysis of whole liver RNA from controls and Atg7 KO for Yap downstream targets. Data indicate row-z scores. **h** qRT-PCR analysis of whole liver RNA from control and Atg7 KO mice at 3 months for Yap downstream targets. Data normalized to *Gapdh* mRNA expression and expression in control mice. Data represent two independent experiments. Mean ± SD. **$P < 0.01$, ***$P < 0.001$ by two tailed *t*-test. **i** Tamoxifen (TAM)-inducible, hepatocyte-specific Atg7 KO (*ERT2-Alb-CRE:Atg7^F/F*) and respective controls were analyzed 7, 14, and 28 days after TAM injection. **j** Immunoblotting of whole liver lysates from control and Atg7 KO mice, 7, 14, and 28 days after tamoxifen injection. **k** Quantitative band densitometry of Yap and β-tubulin immunoblots normalized to Yap/β-tubulin ratios in control mice. **$P < 0.005$ by two tailed *t*-test compared to controls. **l** Band densitometry of P-Yap and Yap immunoblots normalized to P-Yap/Yap ratios in control mice. **$P < 0.005$ and *$P < 0.05$ by two tailed *t*-test compared to controls. Scale bar 100 μm. Insets in **b**, **d**, **e** in the lower right corners correspond to small boxes within the associated lower magnification images

Atg7 floxed mice into the tamoxifen-inducible CRE line driven by the Albumin promoter[46,47] (Fig. 1i). This model allows for time dependent, hepatocyte-specific gene deletion in adult liver. Immunoblotting of whole liver lysates from tamoxifen-inducible, conditional Atg7 KO mice revealed increased total Yap protein within 7 days post tamoxifen injection, concomitant with effective Atg7 deletion and abrogation of autophagy characterized by reduced conversion of LC3-I to LC3-II (Fig. 1j). In contrast, protein levels of the Yap paralog Taz/Wwtr1 were consistently increased from 14 d (Supplementary Fig. 3D). Immunostaining

for Taz/Wwtr1 showed significant staining in non-parenchymal cells rather than hepatocytes (Supplementary Fig. 3E). P62/Sqstm1, a selective autophagy adaptor protein, which accumulates upon inhibition of autophagy[7] and which mediates Nrf2 stabilization via Keap1[21] was increased at 28 days post tamoxifen injection, indicating that the early increase in Yap at d7 is independent of p62/Sqstm1 (Fig. 1j). Quantitative band densitometry showed significantly increased Yap/β-Tubulin and decreased P-Yap/Yap ratios, consistent with Yap activation (Fig. 1k, l).

**Yap is degraded by autophagy.** The Hippo/Yap signaling pathway contains a highly conserved core kinase cassette which, when engaged, phosphorylates and inactivates Yap by βTrCP-mediated proteasomal degradation[38,48]. Yap may also be inactivated by sequestration in the cytoplasm[27,28,32]. Since previous data indicated normal proteasomal activity in autophagy-deficient neurons[49] we hypothesized that Yap is degraded by autophagy. To test our hypothesis, we stably infected the murine hepatocyte line AML12 with lentiviral shRNAs (scrambled or shAtg7[50]) and verified *Atg7* knockdown by qRT-PCR and immunoblotting (Supplementary Fig. 4A, B, C). shAtg7-AML12 cells displayed

increased p62/Sqstm1 typical of autophagy inhibition and increased nuclear Yap localization by immunofluorescence (Fig. 2a), which was also verified by immunoblotting Yap in cytoplasmic and nuclear fractions (Fig. 2b). Gapdh and Nucleoprotein p62 (Nup62) (nuclear Nup62 62 kD, cytoplasmic precursor 61 kD protein)[51] served as markers for the respective fractions. We observed increased nuclear Nup62 in shAtg7 cells despite loading equal amounts of protein per lane, however, increased expression of Nup62 and other nucleoproteins have been described to be cell cycle-regulated with increases of Nup62 from G1 to G2/M phase[52]. Indeed, shAtg7 cells exhibited greater

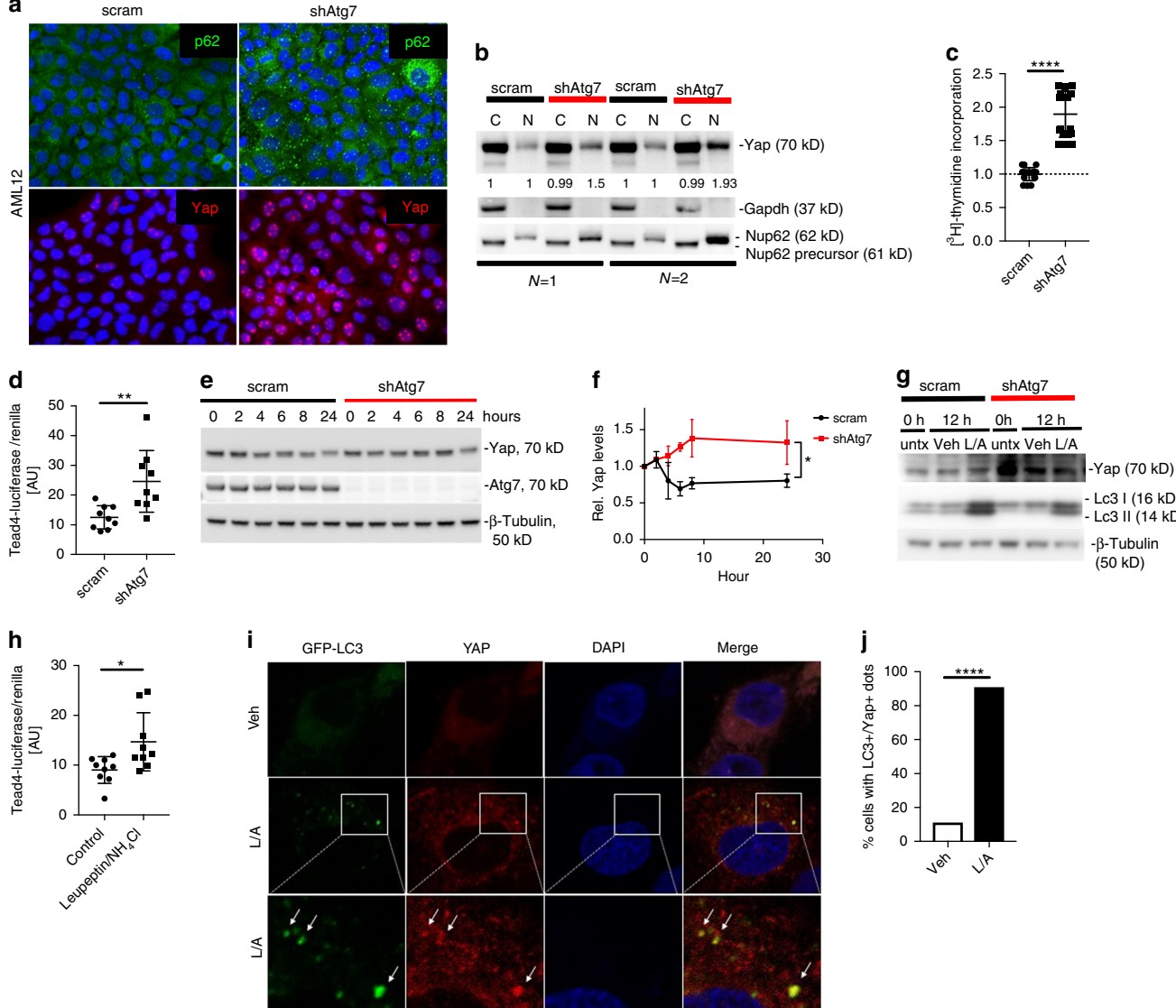

**Fig. 2** Yap is degraded by autophagy. **a** Immunofluorescence analysis of scram- and shAtg7-AML12-infected cells for p62/Sqstm1 and Yap. **b** Subcellular fractionation of cytoplasmic (C) and nuclear fractions (N) from scram- and shAtg7-infected AML12 cells. Immunoblotting for Yap, Gapdh, Nup62. Results representative of four independent experiments. **c** [3H]-Thymidine incorporation assay in scram- and shAtg7-AML12 cells. Mean ± SD. Data from two independent experiments and normalized to scram. Measurements in triplicates. ****$P < 0.0001$ by two tailed *t*-test. **d** Tead4-Luciferase assay of scram- and shAtg7-infected AML12 cells. Data represent three experiments. Mean ± SD. **$P < 0.005$ as determined by two tailed *t*-test. **e** Cycloheximide chase assay of scram- and shAtg7-AML12 infected cells. Immunoblotting of whole cell lysates for Yap, Atg7 and β-Tubulin after cycloheximide incubation. **f** Densitometry analysis for Yap. Data normalized to time 0, mean ± SEM. CHX, cycloheximide. *$P = 0.030$ as determined by paired two tailed *t*-test. **g** Immunoblotting of cell lysates of scram- and shAtg7-AML12 infected cells incubated with Leupeptin/NH4Cl or vehicle only. Immunoblotting for Yap, Lc3 I, II and β-Tubulin. Untx, untreated; Veh, vehicle; L/A, Leupeptin/NH4Cl; h, hours. **h** Tead4-Luciferase assay of AML12 cells incubated with Leupeptin/NH4Cl or vehicle. Data from three independent experiments. Mean ± SD, *$P < 0.05$ as determined by two tailed *t*-test. **i** Confocal immunofluorescence of THLE5B cells transfected with GFP-LC3 and incubated with Leupeptin/NH4Cl or vehicle, staining for YAP and with DAPI for nuclei. **j** Quantitative analysis of cells with LC3+/YAP+ dots. ****$P < 0.0001$ as determined by two tailed *t*-test

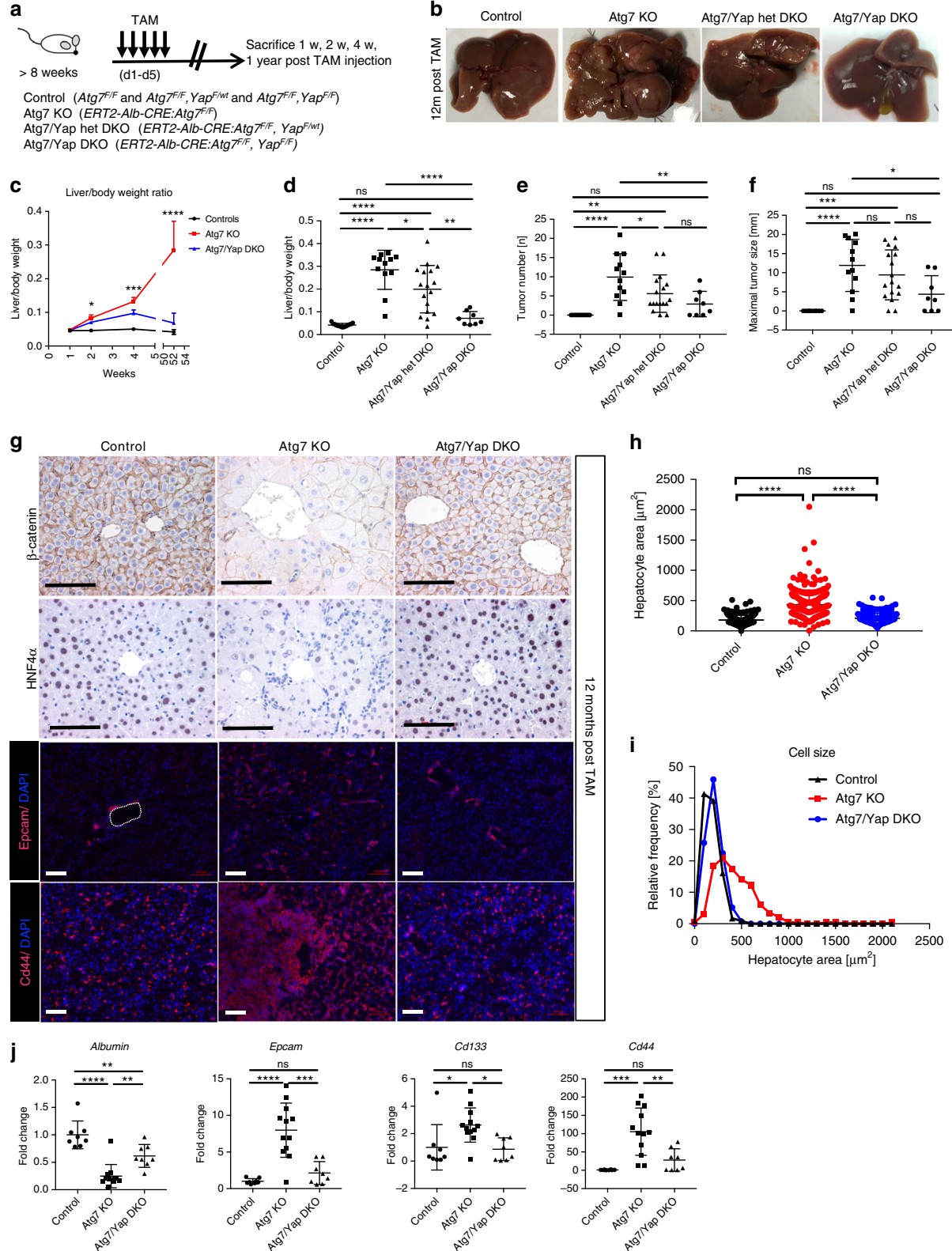

proliferative activity as assessed by [³H]-Thymidine incorporation (Fig. 2c) consistent with nuclear Nup62 increase and Yap-mediated proliferation.

Similarly, shAtg7-AML12 cells transfected with a Tead4-Luciferase reporter[35] displayed significantly more luciferase activity compared to controls, consistent with increased Yap activation (Fig. 2d). By cycloheximide chase assay Yap half-life was increased in shAtg7-AML12 cells (Fig. 2e, f) while proteasomal activity was maintained (Supplementary Fig. 4D). We thus tested if Yap is degraded by autophagy. Leupeptin/NH₄Cl which inhibits lysosomal degradation including autophagy, further increased Yap protein levels in scram-AML12 cells but not in shAtg7-cells (Fig. 2g), confirming that increased Yap levels in shAtg7-cells are a consequence of reduced lysosomal

**Fig. 3** Yap deletion in Atg7 KO mice attenuates hepatomegaly, altered tissue architecture and hepatocarcinogenesis. **a** Control and TAM-inducible, hepatocyte-specific Atg7 KO (ERT2-Alb-CRE:Atg7$^{F/F}$) and Atg7/Yap het DKO (ERT2-Alb-CRE:Atg7 $^{F/F}$, Yap $^{F/wt}$) and Atg7/Yap DKO (ERT2-Alb-CRE:Atg7$^{F/F}$, Yap$^{F/F}$) were analyzed 1 week, 2 weeks, 4 weeks, and 12 months after TAM injection. TAM tamoxifen. N = 15, 12, 17, and 8 animals respectively. **b** Gross liver morphology in control, Atg7 KO and Atg7/Yap DKO after 12 months of TAM injection. **c** Liver/body weight ratio 1 week, 2 weeks, 4 weeks, and 12 months after TAM injection of control, Atg7 KO and Atg7/Yap DKO. Stars indicate significant differences between Atg7 KO and Atg7/Yap DKO at the respective time points by one-way ANOVA and Tukey's HSD. ***P = 0.0001. **d** Liver/body weight ratio in control, Atg7 KO and Atg7 KO/Yap het DKO, Atg7/Yap DKO mice 12 months after TAM induction. ***P < 0.0005. **e** Absolute tumor number per mouse in control, Atg7 KO and Atg7 KO/Yap het DKO, Atg7/Yap DKO mice 12 months after TAM induction. **f** Size of largest tumor size per mouse in control, Atg7 KO and Atg7 KO/Yap het DKO, Atg7/Yap DKO mice 12 months after TAM induction. **g** Immunostaining for β-catenin, HNF4α, Epcam, Cd44 in controls, Atg7 KO and Atg7/Yap DKO 12 months after TAM injection. Scale bar indicates 100 µm. **h** Quantitative analysis of individual hepatocyte areas in controls, Atg7 KO and Atg7/Yap DKO 12 months after TAM injection. n = 238, 242, and 273 cells were analyzed, respectively. **i** Frequency distribution analysis of hepatocyte size in controls, Atg7 KO and Atg7/Yap DKO. **j** qRT-PCR analysis of whole liver mRNA from control, Atg7 KO, Atg7/Yap DKO for Albumin, Epcam, Cd133, and Cd44, normalized to Tbp expression. N = 7, 12, and 8 animals, respectively. Data represent mean ± SD. P-values analyzed by one-way ANOVA and Tukey's HSD. *P < 0.05, **P < 0.005, ***P < 0.001, ****P < 0.0001 unless indicated otherwise. ns, not significant

degradation. Furthermore, Tead4-Luciferase activity was significantly increased in AML12 cells incubated with Leupeptin/NH$_4$Cl compared to cells incubated with vehicle alone (Fig. 2h). In the human hepatocyte line THLE5B[53], YAP protein was also increased upon inhibition with the lysosomal inhibitor Leupeptin/NH$_4$Cl or by the macroautophagy inhibitor 3-methyladenine (Supplementary Fig. 4E). Increased YAP protein levels and Tead4-Luciferase activity indicative of YAP activation was also observed in shATG5-THLE5B cells (Supplementary Fig. 4J, K). Combined, these result indicate that YAP activation results from a general deficiency in the autophagy pathway rather than from loss of ATG7 alone.

To confirm that YAP colocalized with lysosomes, THLE5B cells were transfected with YAP-DsRed and incubated with Lyso®TrackerGreen, which enabled tracking of Yap and lysosomes. Incubation with Leupeptin increased the appearance of lysosomes and the colocalization of YAP and lysosomes, indicating lysosomal degradation of YAP (Supplementary Fig. 4F, G). Similarly, THLE5B cells transfected with YAP-DsRed and GFP-LC3 also showed colocalization of YAP and autophagosomes upon Leupeptin/NH$_4$Cl incubation (Supplementary Fig. 4H,I). Finally, endogenous YAP colocalized with autophagosomes in cells transfected with GFP-LC3 (Fig. 2i, j) upon Leupeptin/NH$_4$Cl incubation, further confirming that YAP is degraded by autophagy.

**Yap deletion attenuates hepatomegaly and hepatocarcinogenesis.** To determine if Yap drives hepatomegaly and hepatocarcinogenesis in Atg7 KO mice, we generated conditional Atg7 and Yap[54] double-floxed mice (Atg7$^{F/F}$, Yap$^{F/F}$) crossed into the tamoxifen-inducible Albumin-CRE line[46]. The use of this system enabled timed and simultaneous hepatocyte-specific knock out of Atg7 and Yap in adult mouse liver. Control (Atg7$^{F/F}$, Yap$^{F/F}$) mice, Atg7 KO (ERT2-Alb-CRE:Atg7$^{F/F}$), Atg7 KO/Yap Het double knock out (ERT2-Alb-CRE:Atg7$^{F/F}$, Yap$^{F/wt}$), and Atg7/Yap DKO (DKO, ERT2-Alb-CRE:Atg7$^{F/F}$,Yap$^{F/F}$) mice were sacrificed 7, 14, 28 days, and 12 months after tamoxifen injection (Fig. 3a). Deletion of Atg7 led to an immediate liver outgrowth apparent at 2 weeks after tamoxifen injection, which was significantly attenuated by homozygous deletion of Yap in Atg7/Yap DKO (Fig. 3c; Supplementary Fig. 5A-E) and complete normalization at 12 months (Fig. 3b–d; Supplementary Fig. 6B, C). Immunostaining and quantification for BrdU$^+$ cells demonstrated significantly increased proliferative activity in Atg7 KO mice at 2 weeks, which was attenuated in Atg7/Yap DKO mice (Supplementary Fig. 5F-H). The attenuation in hepatomegaly was proportionate to the number of Yap alleles deleted (heterozygous or homozygous deletion) at 4 weeks and 12 months following tamoxifen administration (Fig. 3d, Supplementary Fig. 5D).

Moreover, tumor size and number were significantly decreased in Atg7/Yap het DKO mice and Atg7/Yap DKO mice compared to Atg7 KO at 12 months after tamoxifen administration (Fig. 3b, e, f, Supplementary Fig. 6D, E) thereby identifying Yap as a driver of liver growth and hepatocarcinogenesis when autophagy is impaired.

Deletion of Yap in Atg7/Yap DKO mice reduced hepatocyte size (Fig. 3g–i) and significantly improved portal and lobular inflammation, ductular reaction, steatosis, and fibrosis in Atg7/Yap DKO mice (Supplementary Fig. 6F-H). Restored hepatic differentiation was demonstrated by increased immunostaining for HNF4α and qRT-PCR of whole liver RNA for Albumin (Fig. 3g, j) in agreement with previous findings linking Yap overexpression to hepatic dedifferentiation[35,55]. In addition, progenitor cell expansion, which is observed with Yap overexpression[35] was significantly decreased in Atg7/Yap DKO mice as assessed by immunostaining and qRT-PCR for Cd133, Cd44, and Epcam (Fig. 3g, j).

**The p62/Sqstm1-Nrf2 axis is maintained in Atg7/Yap DKO livers.** In previous studies the impairment of hepatic autophagy leads to accumulation of p62/Sqstm1, which then stabilizes the transcription factor Nrf2 (NFE2L2, nuclear factor erythroid-2 related factor 2) via Keap1 (Kelch-like ECH associated protein 1) an adaptor of the ubiquitin-ligase complex[21]. Consequently, liver-specific double knockout of either Atg7 or Atg5 and p62/Sqstm1 or Nrf2 abrogates hepatomegaly and attenuates tumorigenesis in p62/Atg7 DKO mice[16,18]. Moreover, p62/Sqstm1 has been recently identified as an independent oncogene in liver[56]. Therefore, we analyzed if p62/Sqstm1-Nrf2-mediated signaling is maintained in Atg7/Yap DKO mice and conversely, if Yap is activated in Atg7/Nrf2 DKO and Nrf2 KO mice. Immunoblotting of whole liver lysates from Atg7 KO mice, Atg7/Yap KO mice, Atg7/Nrf2 DKO, and Nrf2 KO mice[57] demonstrated increased Nrf2 and p62/Sqstm1 protein in Atg7 KO and Atg7/Yap DKO, but not in Atg7/Nrf2 DKO or Nrf2 KO (Fig. 4a, b). By qRT-PCR analysis of whole liver RNA, Nrf2 downstream targets (Nqo1, Srxn1) were increased in both Atg7 KO and Atg7/Yap DKO mice consistent with preserved Nrf2 activation (Fig. 4c). These results indicate intact p62/Sqstm1-Nrf2 mediated signaling in Atg7/Yap DKO and establish Yap as a separate driver of HCC in autophagy-impaired livers. Our results point to an important role of Nrf2 in p62/Sqstm1 homeostasis and Yap activation as Atg7/Nrf2 DKO mice exhibited Yap levels comparable to control mice and did not have evidence of p62/Sqstm1 increase. On mRNA level, Yap target genes Cyr61 and Areg were significantly decreased in Atg7/Yap DKO compared to Atg7 KO mice, indicating that loss of Yap is sufficient to decrease Hippo pathway target gene expression despite Taz/Wwtr1. This finding is

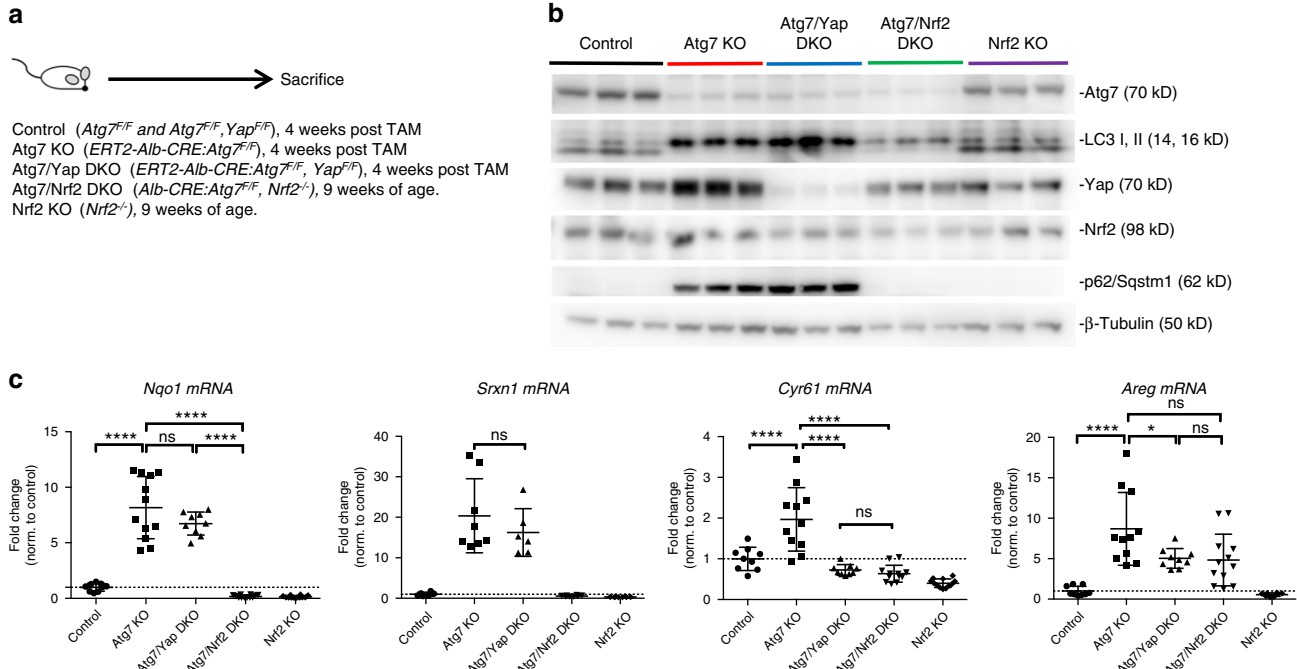

**Fig. 4** Intact Nrf2 signaling in Atg7/Yap DKO mice; Atg7/Nrf2 DKO mice do not have evidence of increased Yap activation. **a** Analysis of control (Atg7$^{F/F}$ andAtg7$^{F/F}$,Yap$^{F/F}$), Atg7 KO (ERT2-Alb-CRE:Atg7$^{F/F}$), Atg7/Yap DKO (ERT2-Alb-CRE:Atg7$^{F/F}$,Yap$^{F/F}$) 4 weeks post TAM injection, Atg7/Nrf2 DKO (Alb-CRE:Atg7$^{F/F}$,Nrf2$^{-/-}$) and Nrf2 KO (Nrf2$^{-/-}$) at 9 weeks of age. **b** Immunoblot analysis of whole liver lysate from control, Atg7 KO, Atg7/Yap DKO, Atg7/Nrf2 DKO, and Nrf2 KO mice. Immunoblotting for Atg7, LC3, Yap, Nrf2, p62/Sqstm1, β-Tubulin. **c** qRT-PCR analysis of whole liver RNA for Nqo, Srxn1, Cyr61, Areg. Data normalized to β-actin expression and fold control. Data from 3 (Cyr61, Nqo, Areg) and 2 (Srxn1) independent experiments, respectively. N = 3–4 animals per group. Mean ± SD. ****$P < 0.0001$ as determined by one-way ANOVA and Tukey's HSD test. ns, not significant. Data represent mean ± SD. P-values analyzed by one-way ANOVA and Tukey's HSD. *$P < 0.05$, **$P < 0.005$, ***$P < 0.001$, ****$P < 0.0001$ unless indicated otherwise

consistent with reports that Yap inactivation likely has a greater impact on cellular physiology than Taz/Wwtr1[58].

**Yap is a potentially druggable driver of HCC in Atg7 KO.** To assess if Yap-dependent proliferation in autophagy-impaired livers (Supplementary Fig. 5G, H) is targetable, we tested the effect of verteporfin, a small molecule Yap-Tead4-inhibitor[59] in culture and in vivo. Verteporfin has also been shown to be safe in phase I/phase II clinical trials[60]. In vivo, treatment of Atg7 KO mice with verteporfin for 21 days significantly decreased the number of Ki67$^+$ nuclei in Atg7KO mice (Fig. 5b, c), which was also associated with reduced expression of the Yap target gene Cyr61 (Fig. 5d). In cultured scram- and shAtg7-AML12 cells, verteporfin significantly decreased proliferative activity (Fig. 5e). Similarly, verteporfin significantly diminished Tead4-Luciferase activity in scram- and shAtg7-AML12 cells (Fig. 5f).

**Atg7-KO gene signature aligns with human NASH and HCC profiles.** To analyze the relevance of our findings to human liver disease and hepatocarcinogenesis, we defined a Atg7-KO gene signature (differentially expressed genes in liver tissues from Atg7-KO mice compared to control mice) and compared it to the transcriptome profile of 72 human non-alcoholic fatty liver disease (NAFLD) liver tissues with known disease severity. GSEA analysis showed significant enrichment of the Atg7-KO gene signature in NAFLD samples that were comprised primarily of samples from advanced NAFLD (F3 or F4), along with enrichment for YAP target genes (Fig. 6a). Similarly, analysis of 374 human HCC transcriptome profiles indicated simultaneous enrichment of the Atg7-KO gene signature and YAP activation in 42.2% (158/374) of HCCs as well (Fig. 6b). Integrative transcriptomic analysis of HCCs has indicated that these tumors may

be classified according to three molecular subclasses, S1, S2, and S3[61]. Interestingly, HCC subclass S1, which is characterized by steatohepatitic HCCs, a histological subtype which arises in NAFLD/NASH[62] and immune cell infiltration[63], was significantly enriched for the Atg7-KO gene signature, whereas subclass S3 had no such alignment with the autophagy-deficient signature. Immunofluorescence analysis of human HCCs with known molecular subclass showed nuclear YAP staining and enhanced cytoplasmic staining for p62/SQSTM1 within the same cells (Fig. 6c), indicating that impaired autophagy flux and concomitant YAP activation occur in human disease.

**Discussion**

By utilizing a regulable hepatocyte-specific promoter to analyze Atg7$^{-/-}$ mice, we have uncovered autophagy as the gatekeeper of hepatic differentiation, growth regulation and carcinogenesis by controlling degradation of Yap. Hepatocyte-specific Atg7$^{-/-}$ mice display enhanced Yap target gene expression and increased Yap protein that drives hepatocyte proliferation, leading to gross hepatomegaly. These abnormalities are significantly attenuated by heterozygous and homozygous loss of Yap allele(s) in Atg7/Yap Het and Atg7/Yap DKO double knockout mice. Thus, Yap is a driver of tissue remodeling and carcinogenesis when autophagy is impaired.

Dysregulation of liver growth regulation in Atg7 KO mice is due to increased cell size (hypertrophy) and Yap-dependent increase in cell number (hyperplasia). Hepatomegaly in Atg7 KO mice was previously attributed to accumulation of non-degraded cell components including proteins, lipids and damaged organelles[17,18,41]. However, our data demonstrate that inhibiting Yap-mediated hepatocyte proliferation and hyperplasia in Atg7/Yap DKO mice can significantly attenuate the increased liver size and significantly decrease tumorigenesis.

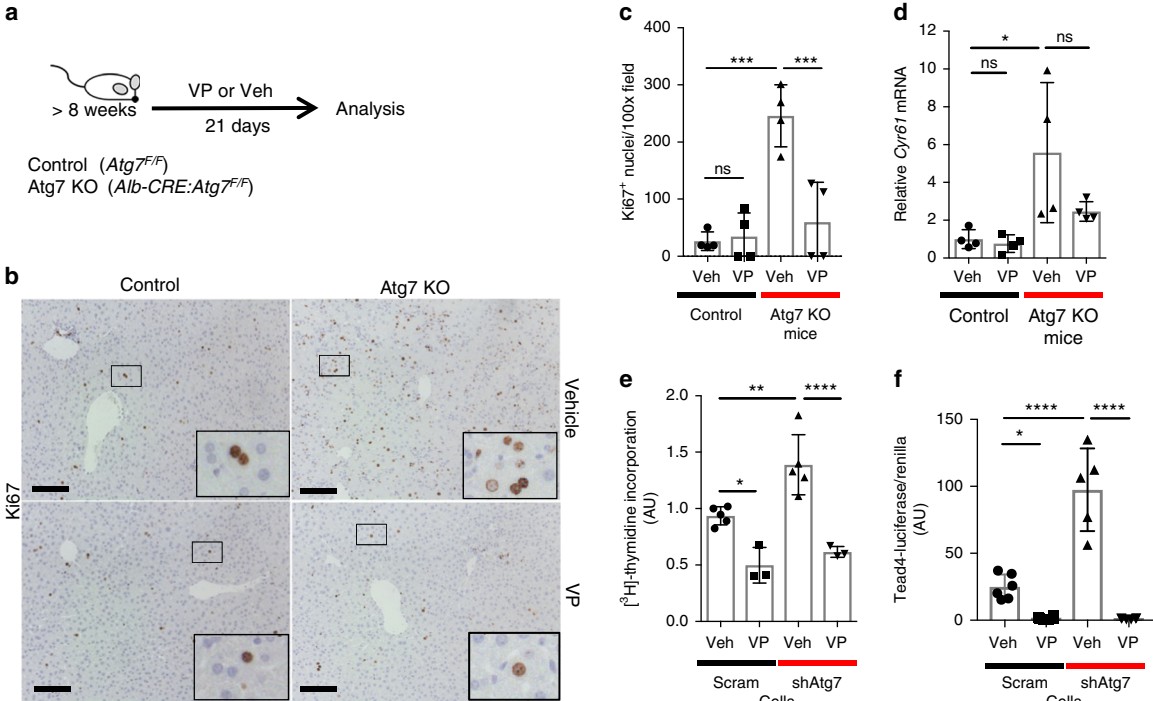

**Fig. 5** Yap-dependent proliferation in livers with impaired autophagy is potentially druggable. **a** Controls ($Atg7^{F/F}$) and Atg7 KO ($Alb$-$CRE$:$Atg7^{F/F}$) mice were treated for 21 days with either Yap/Tead inhibitor verteporfin or vehicle. $N = 4$ per group. VP, verteporfin; Veh, vehicle. **b** Immunostaining for Ki67 in liver sections from control and Atg7 KO mice treated with either verteporfin or vehicle. Scale bar 100 μm. Large inserts are enlargements of small inserts. **c** Quantitation of Ki67+ nuclei per 100x field per mouse treated with either verteporfin or vehicle. Ten 100x fields per mouse were analyzed. **d** qRT-PCR analysis of whole liver RNA from control and Atg7 KO mice, treated with verteporfin or vehicle. $Cyr61$ mRNA expression was normalized to $Gapdh$ expression and then normalized to control animals. Data from two independent experiments. **e** [3H]-thymidine incorporation assay in scram- and shAtg7-AML12 cells after incubation with verteporfin or vehicle. Data from three independent experiments, measurements in triplicates. ****$P = 0.0003$. **f** Tead4-Luciferase assay of scram- and shAtg7-infected AML12 cells after incubation with verteporfin or vehicle. Data from two independent experiments Data represent mean ± SD. $P$-values analyzed by two-way ANOVA and Tukey's HSD. *$P < 0.05$, **$P < 0.01$, ***$P < 0.001$, ****$P < 0.0001$ unless indicated otherwise. ns, not significant; VP, verteporfin; Veh, vehicle

Unlike previous studies[20,21] the dramatic phenotype consequences of autophagy loss in liver occurs independent of p62/Sqstm1-KEAP1-NRF2 signaling in our study. p62/Sqstm1 stabilizes Nrf2 by negatively regulating Keap1, thereby abrogating polyubiquitination and proteasomal degradation of Nrf2[21]; this stabilizes Nrf2 and allows its nuclear translocation. In previous studies, Atg7/p62 ($Alb$-$CRE$:$Atg7^{F/F}$, $p62^{-/-}$) and Atg7/Nrf2 ($Mx1$-$CRE$:$Atg7$ $Atg7^{F/F}$, $Nrf2^{-/-}$) double knock-out mice reportedly rescue the dramatic liver phenotype, associated with a significant decrease in ubiquitinated proteins consistent with improved proteostasis despite inhibition of autophagy[18,21]. In Atg7/p62 double knock-out mice there is decreased tumorigenesis in aged Atg7/p62 DKO animals. Thus, the contribution and relationship of p62/Sqstm1, Nrf2, and Yap in the context of autophagy impairment to liver injury, liver growth regulation and hepatocarcinogenesis remains to be clarified. p62/Sqstm1 is an independent oncogenic driver in AAV-p62/Sqstm1-infected hepatocytes through oncogenic effects of mTOR and Nrf2 signaling. In contrast, Atg7/Yap double knock-out significantly attenuates hepatocarcinogenesis despite increased expression of p62/Sqstm1 and Nrf2, and Nrf2 downstream targets. Thus, Yap is a significant driver of hepatocarcinogenesis when autophagy is impaired. However, tumorigenesis was not completely abrogated in Yap/Atg7 double knock-out mice and the remaining tumorigenesis could be due to p62/Sqstm1-Nrf2 signaling. A contribution of the Yap paralog Taz/Wwtr1 cannot be excluded, however immunostaining demonstrates an increase in Taz/Wwtr1 in non-parenchymal cells, and qPCR analysis showed significant decrease in Hippo downstream targets

$Cyr61$ and $Areg$, indicating efficient suppression of Yap/Taz targets in Atg7/Yap DKO mice.

Autophagy is clearly a tumor-suppressive pathway in our models. In previous studies autophagy has been ascribed a bifunctional role in tumorigenesis—tumor suppressive in the initial phase and tumor promoting in advanced stages[8]. Previously, tumors in hepatic Atg7 KO mice have been described as benign adenomas[17]; however, tumors in ERT2-Alb-CRE:$Atg7^{F/F}$ and Alb-CRE:$Atg7^{F/F}$ mice fulfill characteristics of HCCs. Differences in the genetic background and microbial environment might contribute to the observed phenotypic differences between ours and previous studies, as the microbiome can affect hepatocarcinogenesis, but future studies will need to clarify the underlying mechanisms.

Our results provide an important link between metabolic alterations and mechanisms for Yap activation in hepatocarcinogenesis. With the epidemic of obesity and metabolic syndrome, hyperlipidemia and insulin resistance are frequently observed concomitant to chronic viral hepatitis and NASH[2] and have been shown to decrease hepatic autophagic and lysosomal degradation[40]. Comprehensive integrative molecular analyses (transcriptomic, genomic and epigenomic data) as well as immunological analyses have provided a molecular and immunological classification of HCCs which can be differentiated into a proliferative (S1 and S2) and non-proliferative class[39,64–66]. Of these, S1 subclass is associated with poor prognosis, chromosomal instability and activation of pathways associated with cell proliferation and survival and immune exhaustion[66]. Enrichment

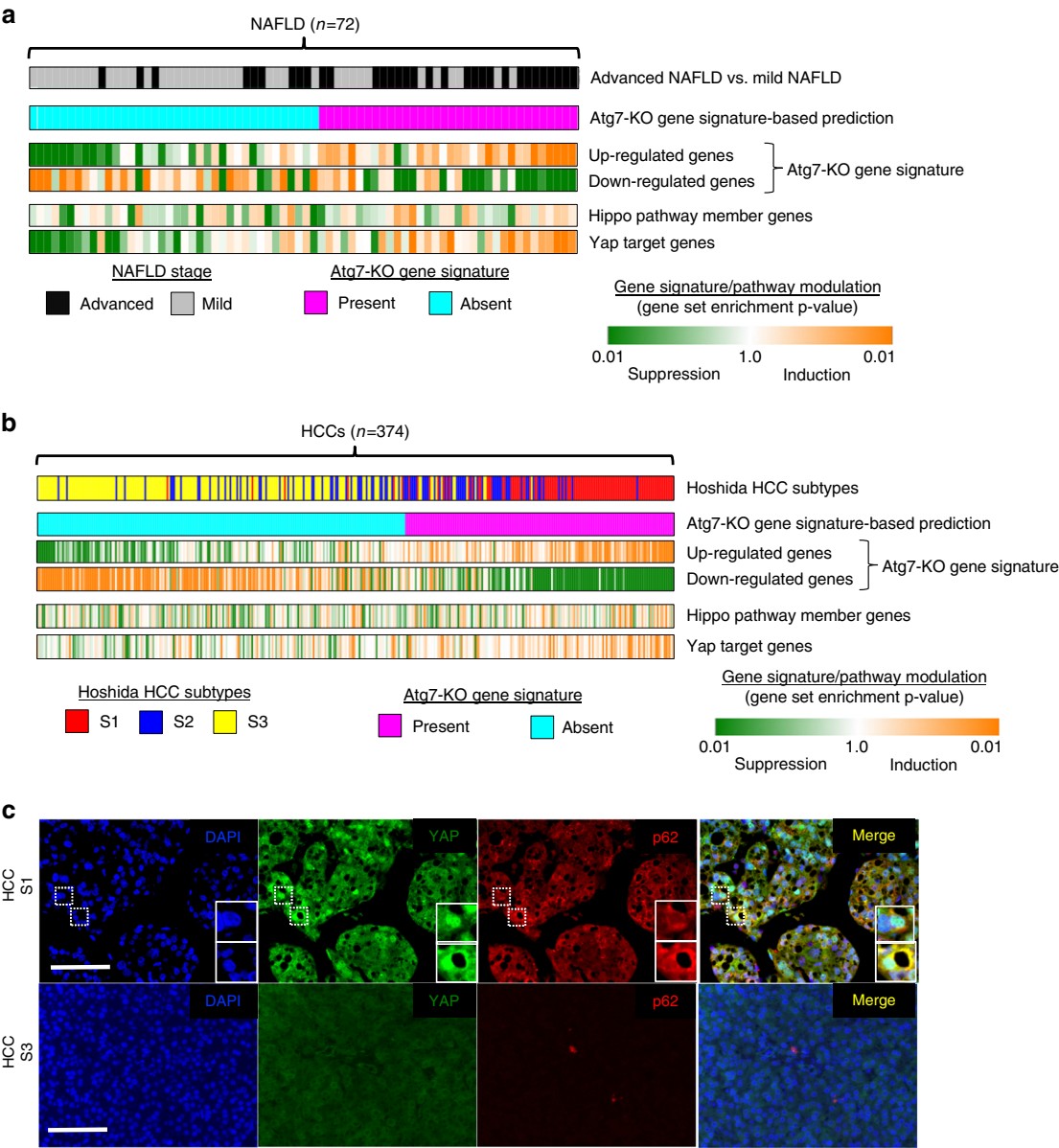

**Fig. 6** Atg7-KO gene signature is enriched in transcriptome profiles of human NAFLD liver tissues and HCCs. **a** Gene set enrichment analysis of transcriptome profiles of NAFLD liver tissues ($n = 74$, advanced (F3, F4) and mild (F0, F1)) with Atg7-KO gene signature and Yap activation signatures. **b** Gene set enrichment analysis of transcriptome profiles of human HCC tissues with Atg7-KO gene signature and Yap activation signatures. **c** Immunofluorescence analysis of human HCC tissue for YAP and P62/SQSTM1. Large insets present magnification of small insets. Scale bar indicates 100 μm

and alignment of the Atg7 KO gene signature in HCCs of the S1 subclass indicates that metabolic pathways of Yap activation might contribute to this subclass.

Our results have uncovered a mechanism of autophagy-dependent degradation of Yap which, when impaired, promotes dedifferentiation, inflammation, fibrosis, and hepatocarcinogenesis. Because Yap activity is strongly implicated in human hepatocarcinogenesis, our studies provide a rationale for chemopreventive or anti-tumorigenic strategies through inhibition of Yap or enhancement of autophagy by drugs such as verteporfin[60] or carbamazepine, which is currently in phase II trial for α1-antitrypsin deficiency (NCT01379469) and has anti-fibrotic effects in murine disease models[67].

## Methods

**Animals**. *Alb-CRE* mice (Jax 003574) were purchased from Jackson Laboratory and crossbred with conditional *Atg7*[19]. *Nrf2*[−/−][21] were crossed with *Alb-CRE:Atg7*[F/F] to generate Atg7/Nrf2 DKO mice. ERT2-Alb-CRE mice[46] crossed with conditional *Atg7*[19] and/or, conditional *Yap* mice[54,21]mice were utilized in this study. Age-matched, male and female mice were used and did not show sex-bias differences. ERT2-Alb-CRE mice (6–8 weeks) and respective controls were given 0.1 mg tamoxifen (Sigma) (diluted in corn oil) for 5 consequential days. Verteporfin (Sigma) diluted in DMSO and injected i.p. (50 mg kg[−1]) every other day for 21 days. All animal studies and procedures were approved by the Animal Care and Use Committee of Icahn School of Medicine at Mount Sinai.

**Immunohistochemistry**. For immunohistochemistry, liver samples were fixed in 10% formalin overnight, transferred to 70% ethanol and processed for paraffin embedding. For immunostaining, samples were deparaffinized and heat induced antigen retrieval performed with either 10 mM sodium-citrate, pH 6 or Tris-EDTA, pH 9 or DAKO target retrieval solution (Dako, S2375) depending on the antigen. Chromogen development after incubation with primary and secondary antibodies was performed using the Dako Envision Kit (Dako). For immunofluorescence studies, AML12 cells, THLE5b cells or primary murine hepatocytes were grown on coverslips, fixed with acetone or 50% acetone/50% methanol for 10 min and rehydrated in PBS. Liver sections from either 4% PFA-perfused livers and sequential sucrose-cryopreservation were analyzed, fixed with acetone, or 50%

acetone/50% methanol or 4% PFA depending on the antigen, washed and incubated with primary antibodies overnight. After washing, cells were incubated with secondary antibodies, embedded and imaged. Primary antibodies and secondary antibodies with dilutions are indicated in Supplementary Table 1. Reticulin staining was performed with the Hito Biotek Kit (HTKCS0102) according to the manufacturer's instructions. For Sirius Red staining, FFPE-slides were deparaffinized, incubated for 15 min in 0.01% fast green in picric acid, washed and then incubated in 0.04% fast green 0.1% Sirius red in picric acid for 45 min. After washing and dehydration, slides were embedded, dried, and imaged. For quantification of Ki67+ hepatocytes, ten fields per mouse were imaged and analyzed. For histological scoring, slides were analyzed by a liver pathologist (MIF) in a blinded manner and graded (0–3 points) analyzing five 200x fields per mouse. Images were analyzed with a Zeiss Axiovision microscope. For confocal live imaging, cells were grown on a glass bottom plate, transfected with Yap-DsRed (addgene 19057) by Lipofectamine 2000 (Roche) and incubated with 100 nM LysoTracker®Green DND-26 (ThermoFisher L7256) in complete medium with 100 µM Leupeptin or vehicle for 2 h prior to imaging. For coimmunofluorescence, cells were transfected with GFP-LC3 via Lipofectamine 2000 (Roche) and incubated with 100 µM Leupeptin/20 mM NH₄Cl or vehicle for 2 h prior to imaging. Microscope was performed at the Microscope CoRE at the Icahn School of Medicine at Mount Sinai.

Morphometric quantitative analysis of cell size and nucleus size from liver sections was performed by ImageJ analysis. Two hundred ten to 270 cells per genotype ($n = 3$) were analyzed and plotted by frequency distribution analysis (Prism 7). Colocalization analysis was performed by ImageJ analysis software (plugin JACoP v2).

**Cytoplasmic and nuclear fractionation.** For cytoplasmic and nuclear fractionation, AML12 cells were grown to 100% confluency, washed twice with ice-cold PBS and incubated with buffer A (10 mM HEPES, 10 mM KCl, 0.1 mM EDTA, 0.1 mM EGTA, pH 7.9, PMSF 50 µg ml⁻¹, Na-Orthovanadate 1 mM, DTT 1 mM, NP-40 (0.6%), HALT-Phosphatase inhibitor (Thermoscientific) 1:1000, complete Protease-inhibitor tablet (Thermoscientific)) on ice for 15 min. Cells were collected and centrifuged for 5 min at $14,000 \times g$. Supernatant (cytoplasmic fraction) was collected and saved. The pellet was processed further for nuclear extraction by washing with buffer A without NP-40, followed by resuspension in buffer C (10 mM HEPES, 0.4 mM NaCl, 1 mM EDTA, 1 mM EGTA, pH 7.9, PMSF 50 µg ml⁻¹, Na-Orthovanadate 1 mM, DTT 1 mM, NP-40 (0.6%), HALT-Phosphatase inhibitor (Thermoscientific) 1:1000, complete Protease-inhibitor tablet (Thermoscientific)) and incubation on ice for 15 min, vortexing every minute. Fraction was centrifuged at $14,000 \times g$ for 8 min and supernatant (nuclear fraction) collected.

**Immunoblotting.** Fifty micrograms of snap frozen liver were homogenized by Qiagen TissueLyser in RIPA buffer containing proteinase inhibitors (Roche complete Mini proteinase inhibitor, 2 tab/10 ml buffer) and 1x HALT phosphatase inhibitors (Thermoscientific, #78428). Lysates were sonicated and then centrifuged for 10 min at $18,800 \times g$. THLE5B and AML12 cells were lysed directly in RIPA buffer containing inhibitors, sonicated and centrifuged. Protein concentration of supernatants were determined by BioRad protein assay (BioRad). For immunoblot analysis, 30–50 µg of protein were analyzed per SDS-PAGE and tank blotting. Antibodies and dilutions used are indicated in Supplementary Table 1. Semi-quantitative band densitometry was performed using ImageJ software. Uncropped scans of blots are provided in the Supplementary Information.

**Hepatocyte isolation and cell culture.** The AML12 cell line was obtained from ATCC (CRL2254) and the THLE5b cells had been obtained from Dr. Yujin Hoshida. The THLE5b cells have been authenticated by short-tandem repeat (STR) profiling. Cell lines were tested for mycoplasma with the Venor GeM Mycoplasma Detection Kit (Sigma, MP0025-1KT) according to the manufacturer's instructions. AML12 cells were cultured in DMEM/F12 medium supplemented with insulin/transferrin/selenium (Invitrogen 51300-044), 40 µg ml⁻¹ Dexamethasone (Sigma D2915) and 10% FBS. AML12 cells were infected with lentiviral shAtg7 or scrambled-shRNA[50] and stable clones selected with puromycin (1.25 µg ml⁻¹). Upon establishing stable clones, cells were maintained without puromycin and stable knock down of Atg7 verified over several passages. THLE5B[68] cells were cultured in DMEM medium supplemented with 10% FBS and infected with shATG5 or scrambled-shRNA (Santa Cruz, sc-41445 and sc-108080, respectively) according to the manufacturer's instructions. Stable clones were selected with puromycin. Primary hepatocytes were isolated by in situ perfusion with Liver Perfusion Medium (Thermo Scientific, 17701038) followed by collagenase B perfusion (Roche, 11088831001, 0.05%) and low grade centrifugation ($50 \times g$). Hepatocytes were plated on collagen-coated plates (BD Biosciences, 354249) at 1 mio cells/6-well and cultured in Williams E medium (10% FBS, 1% L-glutamine, 1% penicillin/streptomycin, hydrocortisone (50 µM), insulin (5 µg ml⁻¹)). Cells were incubated in verteporfin (Sigma SML0534, 10 µg ml⁻¹ final concentration), cycloheximide (Enzo Life Sciences, 50 µM final concentration) or the respective vehicles (polyethylene glycol 300 (PEG 300) or DMSO respectively).

**Luciferase assays.** AML12 and THLE5B cells were transfected at 70% confluency using Mirus TransIT-LT1 transfection reagent and analyzed at 100% confluency to control for cell-contact mediated inhibition of Hippo pathway by Dual-Glo Luciferase Assay (Promega). pRL-Tead4-Luciferase and Renilla control plasmid were gifts of Yimlamai and Camargo[35]. Incubation with verteporfin (10 µg ml⁻¹) or vehicle (PEG300) were carried out for 24 h.

**Cycloheximide chase assay.** AML12 cells were incubated at 90% confluency in complete growth media containing the protein synthesis inhibitor cycloheximide (Enzo Life Sciences) (50 µM final concentration) or vehicle (DMSO). Cells were collected in RIPA buffer (50 mM Tris/HCl, pH 7.4, 150 mM NaCl, 1% Igepal, 0.5% Na-Deoxycholate, 0.1% SDS) containing proteinase inhibitors (ThermoScientific Protease-Inhibitor tablets, 2 tablet/10 ml buffer) at 0, 2, 4, 6, 8, and 24 h after incubation. The lysate was sonicated, centrifuged for 10 min at $18,800 \times g$ and the resulting supernatants analyzed by immunoblotting.

**Proteasomal activity assay.** AML12 cells were incubated in cell lysis buffer (50 mM HEPES, pH 7.5; 5 mM EDTA, 150 mM NaCl and 1% Triton X-100) for 30 min on ice, vortexing every 10 min. Lysates were centrifuged at $20,600 \times g$ for 15 min. Protein concentration of the supernatant was determined via Biorad Protein Assay. Protein lysates (10 µg) were diluted with 10x assay buffer (250 mM HEPES (pH 7.5), 5 mM EDTA, 0.5% NP-40, 0.01% SDS) and samples incubated with proteasome substrate at 37˚C for 90 min. Concentration of proteasome substrates were 100 µM for caspase-like activity (Ac-Gly-Pro-Leu-Asp-AMC, Biomol, AW9560), 10 µM for chymotrypsin-like activity (Proteasome substrate III, Calbiochem, #539142) and 100 µM for trypsin-like activity (Biomol, AW 9785). Release of free hydrolyzed 7-amino-4-methylcoumarin (AMC) groups was measured using an ISS Counter with an excitation filter of 380 nm and an emission filter of 460 nm.

**qRT-PCR and primer sequences.** For qRT-PCR analysis, RNA was extracted from snap frozen liver or cells by Trizol (Ambion) followed by purification using the RNeasy Kit (Qiagen). One to 5 µg RNA were transcribed to cDNA using the RNA to cDNA EcoDry Premix (Clontech). IQ SYBR Green Supermix (Biorad) was used for quantitative PCR on the LightCycler480 System (Roche Diagnostics). Samples were analyzed in triplicates. Data are represented as the relative expression after normalizing to housekeeping genes (*Gapdh*, β*-actin*, *18S, Tbp*) expression, respectively. Primer sequences are listed in Supplementary Table 2.

**Gene expression analysis.** Liver tissue was incubated in RNAlater (Ambion) and RNA extracted by Trizol (Ambion) according to the manufacturer's instructions, and further purified by the RNeasy Kit (Qiagen). RNA from 2 month old, female and male, age-matched *Alb-CRE:Atg7^{F/F}* and *CRE*-negative littermates ($n = 4$ per group) were analyzed on a MouseWG-6 v2.0 Expression BeadChip according to the manufacturer's protocol. Raw scanned data were normalized by using cubic spine algorithm implemented in the GenePattern genomic analysis toolkit (www.broadinstitute.org/genepattern). Probe-level expression data were collapsed into gene-level by calculating the median of multiple probes, and converted to human genes based on an orthologous mapping table provided by the Jackson laboratory (www.informatics.jax.org). Gene expression analysis was determined by R software and expressed as row z-score.

**Bioinformatics and statistical data.** Molecular pathway deregulations were determined in the genome-wide transcriptome dataset by surveying a comprehensive collection of pathway gene sets in Molecular Signature Database (MSigDB, www.broadinstitute.org/mdigdb) and a collection of liver cancer-related gene signatures in literature using Gene Set Enrichment Analysis (GSEA, www.broadinstitute.org/gsea).

Transcriptome profiles of 72 NAFLD-affected liver tissues were obtained from NCBI Gene Expression Omnibus database (www.ncbi.nlm.nih.gov/geo, accession number GSE49541). Genome-wide transcriptome profiles of 374 human HCC tissues were obtained from The Cancer Genome Atlas data portal (https://gdc.cancer.gov). Transcriptomic molecular HCC subtypes we were determined using Nearest Template Prediction (NTP) algorithm as previously described[61,69]. Atg7-KO gene signature was defined as differentially expressed genes in liver tissues from the Atg7-KO mice compared to control mice by *t*-test-based LOOCV validation (four biological replicates for Atg7-ko and control group, respectively). Transcriptional target genes of Hippo pathway (REACTOME_SIGNALING_BY_HIPPO), transcriptional target genes of Yap1 (CORDENONSI_YAP_CONSERVED_SIGNATURE) were obtained from Molecular Signature Database (MSigDB) (www.broadinstitute.org/msigdb). Modulation of the gene sets in each individual sample was determined by modified gene set enrichment analysis[70].

**Human HCC samples.** Formalin-fixed, paraffin-embedded (FFPE) HCC tissues ($n = 8$) were analyzed by immunohistochemistry. The de-identified human HCC samples were obtained from Mount Sinai Biorepository with IRB-approved written informed consents from patients. The study had been approved by the Institutional Review Board of Icahn School of Medicine at Mount Sinai.

## Data availability

The Gene Expression Omnibus accession number for the transcriptome profiles reported in this paper is GSE67676. The Gene Expression Omnibus accession number for 72 NAFLD-affected livers is GSE49541. Genome-wide transcriptome profiles of 374 human HCC tissues were obtained from The Cancer Genome Atlas data portal (https://gdc.cancer.gov).

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

## Acknowledgements

We thank Dr. Dean Yimlamai for critical discussion of the manuscript. Dr. Manjeet Desmukh for HPLC analysis of verteporfin and Jenny Wong for excellent technical assistance. Funding was provided by NIH (RO1DK56621, RO1 AA020709, the US Department of Defense CA150272 and 1P30 CA 196521-01 (S.L.F.) and the German Research Foundation LE 2794/1-1 (Y.A.L), Y.A.L. was also supported in part by grant # UL1TR001866 from the National Center for Advancing Translational Sciences (NCATS), National Institutes of Health (NIH) Clinical and Translational Science Award (CTSA) program. Y.H. received funding from US NIH/NIDDK R01 DK099558, European Union ERC-2014-AdG-671231 HEPCIR, Irma T. Hirschl Trust, U.S. Department of Defense W81XWH-16-1-0363, and Cancer Prevention and Research Institute of Texas RR180016. Funding was provided by the European Union Seventh Framework Program (FP7/2007–2013) grant agreement no. 267248 (L.A.N.), NIH/NIDDK R01 DK103022 (K. N.C.). Fernando Camargo and Dean Yimlamai kindly shared Tead4-Luciferase and renilla constructs. Microscopy was performed at the Microscopy CoRE at the Icahn School of Medicine at Mount Sinai. We thank the Center for Comparative Medicine and Surgery for excellent animal care and Dr. Kevin Kelley for support in vitro rederivation of mice lines.

## Author contributions

Y.A.L., L.A.N., T.L., S.L.F. designed experiments. Y.A.L., L.A.N., M.D.Y., K.M.A., H.C., M.L.B., F.P.N., M.I.F., R.G., B.K., E.K., G.R.J., N.G., N.F., Y.H. performed experiments and data analysis. Y.A.L. and S.L.F. wrote the manuscript. C.P., D.G., X.Y., K.C., Z.Y., M.J.C., A.M.C. designed experiments and provided critical discussion of the manuscript.

## Additional information

**Competing interests:** The authors declare the following competing interests: A.M.C. is co-founder of Selphagy Therapeutics (Boston) and consults for Neuropore (San Diego, CA). S.L.F. was consultant to Vivace Therapeutics. The remaining authors declare no competing interests.

