## [Peer Review File · Nature Communications]

Reviewers' comments:

Reviewer #1 (Liver cancer expert):

The manuscript by Lee and colleagues describes Yap as a key mediator of cell growth in the setting of hepatic ATG7 deficiency. In addition, the authors described that hepatic deletion of ATG7 by two different Cre deleters, Albumin-CRE and Olig1-CRE leads to a similar phenotype. While the latter finding is of limited novelty and relevance, the link between the Hippo and autophagy pathways is novel and interesting. Unfortunately, the manuscript is missing data on how Yap deletion affects hepatocarcinogenesis in mice with hepatic ATG7 deficiency. Also, the effect of Yap deletion on hepatomegaly is only moderate.

1. The manuscript starts with a description of Olig1-CRE-mediated recombination in the liver. While this finding is somewhat surprising, it does not fit well with the overall theme of the manuscript. The authors should move all data on Olig1 to the supplements and only leave the Olig1 data in the regular figures that makes essential for their main topic (which would be pretty much only the confirmation that hepatocyte-specific ablation of ATG7 is sufficient to achieve a phenotype). There are already several sufficiently well established methods to achieve hepatocyte-specific gene ablation besides Olig1CRE (e.g. AAV8-Cre- or Alb-CreERT-mediated). In addition, authors should reorganize their manuscript to start with a clear question/hypothesis instead describing Olig1 phenotypes - and remove most of the first two paragraphs on Olig1 in the results section.. It is confusing for readers to learn about Olig1 and then later realize that the manuscript is not about Olig1 but about the role of Yap in hepatic autophagy deficiency. Moreover, the characterization of ATG7-deficient livers should focus on what is new - everything else should be moved to the supplements.

2. One of the strengths of the manuscript is the demonstration that Yap deletion reduces hepatomegaly. As most of the manuscript is on HCC, it is surprising that a similar ATG7/Yap double knockout was not done for the cancer studies. This data absolutely needs to be included. The presented in vitro data are interesting - but are not a sufficient substitute.

3. The human relevance of the described pathway is not sufficiently investigated. The authors present comparison of similar endpoints - i.e. HCC in mice with hepatic ATG7 deficiency vs human HCC - reveals similarities. The authors need to describe the relevance of the pathway that leads to this endpoint. A complete inhibition of autophagy as in the ATG7 deletion model does not exist in patients. But is there any indication that reduced autophagy increases Yap activation? Or can it be concluded that the total deletion of ATG7 leads to an artificial situation that upregulates Yap but bears no similarities to human hepatocarcinogenesis? Since the effect of Yap deletion in a model of complete ATG7 is only moderate, one wonders how important this mechanism is in more realistic situations, e.g. a moderate 60-80% decrease in autophagy.

4. The direct link between autophagy and the Hippo pathway is not sufficiently demonstrated. It cannot be excluded that the Hippo pathway is important in cell growth regulation but that it acts independently of autophagy, i.e. that deleting Yap reduces hepatomegaly even though it is not functionally linked to autophagy simply because it is an essential regulator of cell growth. In this regards, there almost no increase in Yap in mice with hepatic ATG7 deficiency in Figure 3D despite the authors claiming that there is.

5. The half-life experiments are not convincing. The analysis lacks statistics. As this is a key experiment, the authors should confirm their data by pulse-chase analysis.

Minor comments

1. In view of previous publications, there should be more thorough confirmation that tumors in the

ATG7-deficient livers are indeed hepatocellular carcinomas.

Reviewer #2 (Autophagy expert):

This manuscript investigates the role of the core autophagy regulator, ATG7 in the liver. The authors demonstrate the unexpected expression of an established oligodendrocyte promoter (Olig1)-driven Cre recombinase specifically in hepatocytes; as a result, the hepatocyte-specific deletion of ATG7 results in hepatomegaly and inflammatory fatty liver disease. These results confirm previous work using Albumin-Cre mediated deletion of ATG7, but now establish that these phenotypes arise specifically from hepatocyte deletion. Moreover, using expression array and GSEA, the authors demonstrate the activation of a Yap signature in ATG7 deleted hepatocytes. Based on this signature and genetic studies, the authors propose that the activation of Yap is responsible for the hepatomegaly in autophagy-deficient livers. Moreover, this Yap signature is present in a poor prognosis subclass of human HCC. Although there is a potential concomitant increase in protein levels of both p62/SQSTM1 (suggestive of reduced autophagy) and YAP via IHC, this human tissue study involves a very small sample size and the conclusions appear quite tenuous.

Overall, the findings will be of interest in the field of autophagy and cancer. The genetics and transcriptional profiling both support a role for Hippo/YAP signaling in the pathology of autophagy-deficient liver although the contribution of this pathway in driving hepatomegaly appears to be partial. Although the data appears overall sound, several results need better quantification, additional data-points or larger sample sizes. Also, it is unclear whether YAP is a direct or indirect target of autophagy deficiency.

1) Although the nuclear staining and the transcriptional profile both support that YAP is activated, the data in Figure 3c-d require better quantification and statistical analysis. The stabilization of YAP in ATG7 null hepatocytes is modest and variable, based on the blot provided. Further quantification is necessary to support this conclusion. Moreover, the P-YAP/YAP ratio is not significantly reduced based on the quantification shown in Fig. 3c; in addition, it is unclear whether the Hippo signaling pathway is suppressed based on the data provided. The blots for these markers in Fig 3d are unconvincing, and it is uncertain whether the PLATS/LATs and PMOB/MOB ratios in Fig 3c are statistically significant.

2) The in vitro studies in Fig 4a-d are interesting and should be repeated with additional ATG knockdowns. As it stands, the data provided in the paper are uniquely focused on ATG7 both in vitro and in vivo; the current standard in the field is to evaluate multiple ATGs in order to establish a general role for the autophagy pathway. These in vitro studies are the best way forward to establish a general role for autophagy in the control of the Hippo pathway in hepatocyte as opposed to an ATG7-specific phenotype.

3) The studies in Fig 4d indicate that ATG7 knockdown leads to increased YAP half-life, but it is unclear whether this a direct or indirect effect of autophagy deficiency. Can the authors provide some functional insight into whether YAP is a direct substrate of autophagy or whether the turnover is secondary to impaired proteosomal activity in ATG deficient cells?

4) As pointed out by the authors, the double ATG7/YAP knockout, the reduction of liver size is only partial; indeed the double KO still appears significantly enlarged when compared to the control. This suggests YAP is a contributor, but not necessarily a driver of the hepatomegaly phenotype in ATG7 KO liver; this data contrasts with previous studies of p62 and NRF2 deletion in ATG7 deficient liver. The authors show p62 accumulation in the DKO liver and propose a continued role for this pathway in promoting liver size. To further support this possibility, the authors should

assess whether NRF2 targets still activated in the DKO liver.

5) Was the pathological quantifications in the paper, such as in Figure 1 and in Supplemental Fig 2, performed in a blinded manner? It is not clear from the figure legends and methods. Whether or not this is the case should be explicitly indicated.

6) The "correlation" between p62 levels and YAP in human HCC in Figure 7 is interesting, but the sample size is extremely small, with only n=2-4 per sub-class. Given the provocative nature of this result, a larger sample size would be much more reassuring. Lastly, it does not appear that YAP is nuclear in these samples, which argues against a functional role for YAP activation in this sub-class. Overall, the conclusions in this figure are quite tenuous.

Reviewer #3 (Hippo signalling expert):

Summary: Here, Lee et al. present data using the conditional ATG7 knockout mouse using a number of different Cre recombinase lines. The ATG7 liver knockout has been previously reported utilizing the MX-1 Cre (Komatsu M et al. (2010) NCB 12, 3:213-224) which displays a liver overgrowth phenotype.

Lee et al. distinguish their work in this paper by emphasizing that hepatocyte-specific loss of ATG7 accounts for a majority of the liver overgrowth phenotype. Furthermore, this overgrowth phenotype is associated with changes in Hippo signaling by decreased degradation of Yap which has been previously described to lead to liver overgrowth and cancer.

My general enthusiasm for this work is dampened by the rigor with which the data is presented. In brief:

- The Olig1-cre line is insufficiently characterized to be considered hepatocyte-specific.
- A clear hypothesis and mechanism for how Yap level/activity increases in the Atg7 knockout model is not presented.
- There is an absence of comparison with regards to Atg7 KO and Yap liver overexpression models.
- The use and presentation of statistics throughout the paper is inconsistent. There are times when it is clear and at other times, notation of the use of a statistical test is absent, although reading the text would suggest that a solid conclusion could be made.

In general, the idea that increased Yap levels and activity contributes to overgrowth in the Atg7 KO model is novel and is a substantial contribution to the field. The genetic and pharmacologic data in the various mouse models for a role in Hippo signaling is strong, but the cell biology data does not elucidate a mechanism of how this is accomplished. Further work in this area should be done to make this manuscript acceptable.

I will highlight specific points to be addressed below.

- Olig1-Cre is a novel hepatocyte specific promoter in liver

The authors present several examples of liver cells in a reporter mouse to argue that the Olig1-Cre line is hepatocyte specific. "Representative" single images of various liver cells types is insufficient to fully establish if this Cre line is truly hepatocyte specific. Often, cre lines are not completely penetrant in the intended cell type, nor do they have absolute fidelity. Characterizing the timing of when Olig1 becomes active in the liver is also important for the reader to imagine when the described phenotype may begin.

Preferentially, quantifying the proportion of hepatocytes vs non-parenchymal cell labeling should

be performed in the Olig1 cre line. A comparison between hepatocytes and cholangiocytes should also be done, especially since a known deficiency of the Alb-cre line is that there is some early cholangiocyte labeling in that model. This may also be true in the Olig1-cre line. Commonly, quantification is done by manually counting dual or triple labeled cells using IHC or automated counting utilizing FACS from dissociated and labeled tissues (See Yanger et al. (2013), *Gene & Development*. 27: 719-24.).

- "Hepatocytes were hypertrophic and hyperplastic...composed of at least 2-3 cells instead of 1 cell (Fig 1d)..."

The resolution of the presented image is difficult to pictorially appreciate this distinction. A reticulin stain of the control and ATG7 knockout should highlight these described differences for the reader. Also, how old is the presented mouse in this example?

- Figure 2d - The ATG7 KO vs Yap signature GSEA of FDR 0.145 is not significant, contrary to the statement by the authors that "GSEA analysis uncovered enrichment of Yap activation gene signatures...". The MST1/2 KO signature is strongly significant (FDR 0.0) and would support that Hippo signaling is inactivated, but there are also non-Hippo pathway substrates that are direct downstream targets of the MST kinases. It is possible (although unlikely) that non-Hippo pathway targets of MST cause the GSEA to align perfectly with the ATG7 KO. The Yap signature assayed in this figure is derived from breast cancer cells and could explain why the FDR as compared to liver cells is not significant.

Since the authors argue that Yap overexpression is a major reason why the liver is enlarged in the ATG7 knockout model, a GSEA compared against available liver specific Yap overexpression models such as Dong et al. (2007) or Yimlamai et al. (2014) should be presented and would be more relevant.

- Figure 2e - "Afp, BIRC5, Gli2, ITGB2" - Have not been shown to be a direct targets of Yap/TEAD by DNA binding assay. These genes have repeatedly been positively associated with Yap activity, but cannot be strictly referred at Hippo target genes. Please provide direct references if this is not correct. Alternatively, other genes such as Jag1 and Notch2 have been shown to be Yap/TEAD targets in the liver and the authors could consider assaying these genes.

"Areg2" is not a gene in GenBank. Areg (amphiregulin) is a reported target gene for Hippo signaling. Please clarify.

- Figure 2f - Please clearly identify the Hippo target genes that are upregulated during ATG7 knockout. As stated above, some genes assayed (ie. Birc5, Afp, Itgb2...) are not strictly Hippo target genes as stated in the text.

- "primary hepatocytes isolated from Olig1-Cre:Atg7f/f mice..." - Details of this procedure could not be found in the methods and would be important to future investigators should they like to repeat these experiments.

Figure 3d - This set of blots is not visually consistent with the author's statements that Yap and Yap activity (as evidenced by pYap/Yap) is increased in the Olig1-Cre/ATG7 KO model. There is no loading control in the first set of blots for comparison. Is the reader supposed to refer to the β -tubulin loading control in the second set of blots for reference? If so, then why does the Taz lane appear so significantly different between these two sets of blots.

To this reviewer, the Yap levels in controls versus Olig1-cre/ATG7 line appear very similar. The quantification in Figure 3c of Yap activity (p-Yap/Yap) does not show significant difference which does not support the idea of increased Yap activity in this model. In comparison, Taz appears to be markedly enriched in this set of blots as well as in the second presented set. The authors have

suggested that Taz does not operate in hepatocytes by their IHC (Fig 3a), so we are left wondering if there is reduced Hippo signaling in the hepatocyte compartment as stated in the text.

Please provide darker exposures of P-Mst1/2 and Mob blots.

- Supplementary Figure 3e/f - No statistical tests are presented to help the reader evaluate the impact of Cyr61/Afp differences, so the statement that "Cyr61 and Afp demonstrated markedly increased expression in Olig1-cre...Yap activation is a specific effect of autophagy impairment" cannot be seriously considered. The data from this figure suggest that BDL and CCl4 injury also activate Yap activity, in some cases to a similar degree as the Olig1-cre Atg7 knockout. Also, other authors have shown some data to that in the presented models (ie. PHx, Wang C et al 2012./Wu H et al 2013.) Yap activation is occurring. It may be a truer statement that Atg7 knockout more potently activates Yap activity.

- "Loss of Atg7 drives Yap-mediated cell proliferation..." - This section of the paper likely is one of the most novel and important portions of the manuscript, but is not well assembled. As a reader, I am not certain how Yap levels/activity increase, only that it does and would be consistent with the general premise of the paper. The general impression that I have, is that dysregulation in autophagy leads to impairment in proteosomal degradation of Yap. This hypothesis, nor an alternative hypothesis of how Yap accumulates in this model is not clearly stated.

"...proteasome, but other inactivating mechanisms have been described." - What are these "other inactivating mechanisms"? In a literature search I only identified various means of ubiquitination that lead to Yap degradation.

"...which revealed a significant increase in Yap protein stability (Fig 4d, e)..." - The Yap blots presented for controls and shAtg7 do not appear significantly different. The quantification of these Yap blots in 4e appear different, but there is no indication that there is a statistically significant difference (4h-8h).

"...proteasomal activity was not impaired...it was increased in shAtg7-AML12 cells...similar to features of primary hepatocytes..." - Sup Fig 5 shows a significant increase in proteosomal activity in shATG7 cells, but no difference or even a decrease in such activity in primary hepatocytes. This statement is not consistent with the presented data.

- "...a specific Yap target and Areg2..." - Areg2 does not exist. See above.

Reviewers' comments:

Reviewer #1 (Liver cancer expert):

The manuscript by Lee and colleagues describes Yap as a key mediator of cell growth in the setting of hepatic ATG7 deficiency. In addition, the authors described that hepatic deletion of ATG7 by two different Cre deleters, Albumin-CRE and Olig1-CRE leads to a similar phenotype. While the latter finding is of limited novelty and relevance, the link between the Hippo and autophagy pathways is novel and interesting. Unfortunately, the manuscript is missing data on how Yap deletion affects hepatocarcinogenesis in mice with hepatic ATG7 deficiency. Also, the effect of Yap deletion on hepatomegaly is only moderate.

1. The manuscript starts with a description of Olig1-CRE-mediated recombination in the liver. While this finding is somewhat surprising, it does not fit well with the overall theme of the manuscript. The authors should move all data on Olig1 to the supplements and only leave the Olig1 data in the regular figures that makes essential for their main topic (which would be pretty much only the confirmation that hepatocyte-specific ablation of ATG7 is sufficient to achieve a phenotype). There are already several sufficiently well established methods to achieve hepatocyte-specific gene ablation besides Olig1CRE (e.g. AAV8-Cre- or Alb-CreERT-mediated). In addition, authors should reorganize their manuscript to start with a clear question/hypothesis instead describing Olig1 phenotypes - and remove most of the first two paragraphs on Olig1 in the results section. It is confusing for readers to learn about Olig1 and then later realize that the manuscript is not about Olig1 but about the role of Yap in hepatic autophagy deficiency. Moreover, the characterization of ATG7-deficient livers should focus on what is new - everything else should be moved to the supplements.

We thank the reviewers for their comments. We agree that the information on Olig-1 was distracting, and have thus removed these studies, and instead only used Albumin-Cre drivers, including a new model with a tamoxifen-inducible Albumin-CRE promoter.

2. One of the strengths of the manuscript is the demonstration that Yap deletion reduces hepatomegaly. As most of the manuscript is on HCC, it is surprising that a similar ATG7/Yap double knockout was not done for the cancer studies. This data absolutely needs to be included. The presented in vitro data are interesting - but are not a sufficient substitute.

We agree and have now included comprehensive data analyzing the phenotype of Atg7/Yap double knock-outs with a homozygote knock-out of Yap rather than only heterozygote knock-outs. These Atg7/Yap DKO mice have significantly decreased liver size, decreased hepatocarcinogenesis and improved differentiation.

3. The human relevance of the described pathway is not sufficiently investigated. The authors present comparison of similar endpoints - i.e. HCC in mice with hepatic ATG7 deficiency vs human HCC - reveals similarities. The authors need to describe the relevance of the pathway that leads to this endpoint. A complete inhibition of autophagy as in the ATG7 deletion model does not exist in patients. But is there any indication that reduced autophagy increases Yap activation? Or can it be concluded that the total deletion of ATG7 leads to an artificial situation that upregulates Yap but bears no similarities to human hepatocarcinogenesis? Since the effect of Yap deletion in a model of complete ATG7 is only moderate, one wonders how important this mechanism is in more realistic situations, e.g. a moderate 60-80% decrease in autophagy.

We agree that complete inhibition of autophagy is not observed in humans. Cell culture studies from shAtg7-AML12 cells, which have approximately 85% knock down of both Atg7 mRNA and protein (Suppl. Fig. 3A, B, C) showed significant increase in nuclear Yap and Tead4-Luciferase activity (Fig. 2A, B, C).

To further determine if this pathway has relevance to human disease, we have performed GSEA analysis on gene expression data from livers of 72 NAFLD patients and also 374 human HCCs. We found significant enrichment for an Atg7-KO gene signature, that was defined by differentially expressed genes from Atg7 KO (Albumin-CRE/Atg7^{F/F}) to controls. These are now described in the Results and Fig. 6. These findings correlated also with enrichment of for gene signatures associated with Yap target gene activation in the NASH samples, thus underscoring potential clinical relevance.

Decreased autophagic flux contributes to chronic liver disease in particular associated with NASH through a number of defects, including hyperlipidemia, hyperinsulinemia and aging¹. NASH in turn is associated with a rising risk of HCC, even in the absence of cirrhosis². HCC is multifactorial and characterized by decreased autophagy (reviewed in³). Since no somatic or germline mutations in the Yap signaling cascade have been identified in HCC to account for its accumulation, the decline in autophagy associated with NASH provides a permissive state in which decreased autophagy leads to Yap accumulation. Although the conditions of complete loss of autophagy in a mouse model are indeed extreme, the cumulative effect of declining autophagy with aging, and in a chronically injured human liver over years (rather than weeks in mice), combined with other convergent genetic and cellular defects precipitate a state in which Yap activation and cellular dedifferentiation further promote neoplasia.

4. The direct link between autophagy and the Hippo pathway is not sufficiently demonstrated. It cannot be excluded that the Hippo pathway is important in cell growth regulation but that it acts independently of autophagy, i.e. that deleting Yap reduces hepatomegaly even though it is not functionally linked to autophagy simply because it is an essential regulator of cell growth. In this regard, there almost no increase in Yap in mice with hepatic ATG7 deficiency in Figure 3D despite the authors claiming that there is.

In this new resubmission, we provide comprehensive data from cultured murine and human cell lines showing that Yap is degraded by autophagy, thus linking the Hippo pathway and autophagy directly (Fig. 2, Suppl. Fig. 3).

By now including studies in mice with a tamoxifen-inducible promoter (Albumin-CRE/Atg7^{F/F}) analysis of liver lysates at 7d, 14d and 21 days post-tamoxifen injection, we clearly show increased Yap protein levels compared to controls, which are present beginning at 7d post tamoxifen injection (Fig. 1J).

5. The half-life experiments are not convincing. The analysis lacks statistics. As this is a key experiment, the authors should confirm their data by pulse-chase analysis.

As requested, we have included statistics and additional data indicating that Yap is degraded by autophagy, thus confirming our hypothesis by multiple approaches. Although isotope-based amino acid pulse chase studies has some advantages over cycloheximide chase, both methods are considered equally viable strategies to assess protein half-life^{4,5}.

Minor comments

1. In view of previous publications, there should be more thorough confirmation that tumors in the ATG7-deficient livers are indeed hepatocellular carcinomas.

We understand the concern of the reviewers and thus have performed the following methods to verify diagnosis of HCC of the tumors in Atg7-deficient livers (ERT2-Albumin-Cre/Atg7^{F/F} and Albumin-CRE/Atg7^{F/F}).

1. Reticulin staining – loss of reticulin staining is typical for HCCs (Suppl. Fig. 2D)
2. Gst1- staining – diffuse staining is typical for HCCs (Suppl. Fig. 2D)
3. Blinded evaluation by an expert liver pathologist (M. Isabel Fiel with > 20 year of experience)
4. Pathognomonic histological changes such as glandular transformation with bile accumulation (Suppl. Fig. 2E).

Reviewer #2 (Autophagy expert):

This manuscript investigates the role of the core autophagy regulator, ATG7 in the liver. The authors demonstrate the unexpected expression of an established oligodendrocyte promoter (Olig1)-driven Cre recombinase specifically in hepatocytes; as a result, the hepatocyte-specific deletion of ATG7 results in hepatomegaly and inflammatory fatty liver disease. These results confirm previous work using Albumin-Cre mediated deletion of ATG7, but now establish that these phenotypes arise specifically from hepatocyte deletion. Moreover, using expression array and GSEA, the authors demonstrate the activation of a Yap signature in ATG7 deleted hepatocytes. Based on this signature and genetic studies, the authors propose that the activation of Yap is responsible for the hepatomegaly in autophagy-deficient livers. Moreover, this Yap signature is present in a poor prognosis subclass of human HCC. Although there is a potential concomitant increase in protein levels of both p62/SQSTM1 (suggestive of reduced autophagy) and YAP via IHC, this human tissue study involves a very small sample size and the conclusions appear quite tenuous.

Overall, the findings will be of interest in the field of autophagy and cancer. The genetics and transcriptional profiling both support a role for Hippo/YAP signaling in the pathology of autophagy-deficient liver although the contribution of this pathway in driving hepatomegaly

appears to be partial. Although the data appears overall sound, several results need better quantification, additional data-points or larger sample sizes. Also, it is unclear whether YAP is a direct or indirect target of autophagy deficiency.

1) Although the nuclear staining and the transcriptional profile both support that YAP is activated, the data in Figure 3c-d require better quantification and statistical analysis. The stabilization of YAP in ATG7 null hepatocytes is modest and variable, based on the blot provided. Further quantification is necessary to support this conclusion. Moreover, the P- YAP/YAP ratio is not significantly reduced based on the quantification shown in Fig. 3c; in addition, it is unclear whether the Hippo signaling pathway is suppressed based on the data provided. The blots for these markers in Fig 3d are unconvincing, and it is uncertain whether the PLATS/LATs and PMOB/MOB ratios in Fig 3c are statistically significant.

In this revised submission, we provide additional data that Yap is activated in cell lines following knockdown of Atg7, or when cultured with autophagy inhibitors. We provide data that Yap is degraded by autophagy and demonstrate that increased levels of Yap may be observed as soon as 7 days after Atg7 deletion in a mouse model with a tamoxifen-inducible, hepatocyte specific promoter (Fig. 1J). As there are also phosphorylation-independent means of Yap inactivation (sequestration with Dystrophin or Angiomotin), we have not further pursued characterization of the Hippo core kinase cassette.

2) The in vitro studies in Fig 4a-d are interesting and should be repeated with additional ATG knockdowns. As it stands, the data provided in the paper are uniquely focused on ATG7 both in vitro and in vivo; the current standard in the field is to evaluate multiple ATGs in order to establish a general role for the autophagy pathway. These in vitro studies are the best way forward to establish a general role for autophagy in the control of the Hippo pathway in hepatocyte as opposed to an ATG7-specific phenotype.

We replicated our culture experiments with autophagy inhibitors and found evidence of Yap activation as well (Fig. 2F, G, H, Suppl. Fig. 3E). While genetic knock downs of other Atg proteins would broaden the data, we hope this approach is acceptable to the reviewers.

3) The studies in Fig 4 indicate that ATG7 knockdown leads to increased YAP half-life, but it is unclear whether this a direct or indirect effect of autophagy deficiency. Can the authors provide some functional insight into whether YAP is a direct substrate of autophagy or whether the turnover is secondary to impaired proteosomal activity in ATG deficient cells?

We have generated data supporting the conclusion that Yap is degraded by autophagy (Fig. 2B, C, E, F, G, H, Suppl. Fig. 3E, F). Proteasomal activity in shAtg7 cells was comparable to or increased when compared to control cells (Suppl. Fig. 3D).

4) As pointed out by the authors, the double ATG7/YAP knockout, the reduction of liver size is only partial; indeed the double KO still appears significantly enlarged when compared to the control. This suggests YAP is a contributor, but not necessarily a driver of the hepatomegaly phenotype in ATG7 KO liver; this data contrasts with previous studies of p62 and NRF2 deletion in ATG7 deficient liver. The authors show p62 accumulation in the DKO liver and propose a continued role for this pathway in promoting liver size. To further support this possibility, the authors should assess whether NRF2 targets still activated in the DKO liver.

We have now included comprehensive new data from our Yap homozygous double knock out experiments rather than only heterozygote knock outs which demonstrate that Yap is both a

driver of hepatomegaly and hepatocarcinogenesis in Atg7 deficient livers (Fig. 3, A-F, Suppl. Fig. 4, Suppl. Fig. 5). Indeed, after long term deletion of Yap (up to 12 months after Yap and Atg7 deletion) DKO mice had a normalized liver to body weight ratio and improved liver injury and fibrosis.

To clarify how our results align with previously published literature and to assess the importance of the p62/Sqstm1-Nrf2 axis in liver growth and hepatocarcinogenesis, we have performed immunoblot of whole liver lysates and have analyzed whole liver RNA from Atg7 KO, Atg7/Yap DKO and Atg7/Nrf2 DKO mice (Fig. 4). Interestingly, Atg7/Yap DKO mice have increased levels of p62/Sqstm1 and Nrf2, as well as increased expression of Nrf2 target genes. However, despite preserved p62/Sqstm1-Nrf2 signaling, Yap deletion significantly reduces liver size and HCC, thus identifying Yap as an important contributor independent from the p62/Sqstm1-Nrf2 axis. Moreover, immunoblotting of whole liver lysates from tamoxifen-inducible Atg7 deletion shows increased Yap activation levels at d7 (Fig. 1J) while p62/Sqstm1 levels only increase at d28, indicating that p62/Sqstm1 is not necessary for Yap activation. In contrast, Atg7/Nrf2 DKO mice had no evidence of increased p62/Sqstm1, Nrf2 and Yap, indicating that Nrf2 has a separate, important role in proteostasis of cells.

5) Was the pathological quantifications in the paper, such as in Figure 1 and in Supplemental Fig 2, performed in a blinded manner? It is not clear from the figure legends and methods. Whether or not this is the case should be explicitly indicated.

All pathological analyses were performed in a blinded manner. We have specified this in the methods section as requested.

6) The "correlation" between p62 levels and YAP in human HCC in Figure 7 is interesting, but the sample size is extremely small, with only n=2-4 per sub-class. Given the provocative nature of this result, a larger sample size would be much more reassuring. Lastly, it does not appear that YAP is nuclear in these samples, which argues against a functional role for YAP activation in this sub-class. Overall, the conclusions in this figure are quite tenuous.

We agree that the sample size of human HCCs analyzed is very small and have included pictures from other samples showing increased nuclear YAP staining and p62/SQSTM1 (Fig. 6C). However, these liver tissues are scarce and quite expensive to analyze (since both gene expression analysis and staining needs to be performed). We were thus limited in the sample size available.

To further determine if this pathway has relevance to human disease, we have performed GSEA analysis on gene expression data from livers of 72 NAFLD patients and 374 human HCCs (Fig. 6A, B). We found significant enrichment for an Atg7-KO gene signature, that was defined by differentially expressed genes from Atg7 KO (Albumin-CRE/Atg7^{F/F}) to controls. This correlated also with enrichment for gene signatures associated with Yap target gene activation in the same samples, thus underscoring potential clinical relevance.

Reviewer #3 (Hippo signalling expert):

Summary: Here, Lee et al. present data using the conditional ATG7 knockout mouse using a number of different Cre recombinase lines. The ATG7 liver knockout has been previously reported utilizing the MX-1 Cre (Komatsu M et al. (2010) NCB 12, 3:213-224) which displays a liver overgrowth phenotype.

Lee et al. distinguish their work in this paper by emphasizing that hepatocyte-specific loss of

ATG7 accounts for a majority of the liver overgrowth phenotype. Furthermore, this overgrowth phenotype is associated with changes in Hippo signaling by decreased degradation of Yap which has been previously described to lead to liver overgrowth and cancer.

My general enthusiasm for this work is dampened by the rigor with which the data is presented. In brief:

- The Olig1-cre line is insufficiently characterized to be considered hepatocyte-specific.
- A clear hypothesis and mechanism for how Yap level/activity increases in the Atg7 knockout model is not presented.
- There is an absence of comparison with regards to Atg7 KO and Yap liver overexpression models.
- The use and presentation of statistics throughout the paper is inconsistent. There are times when it is clear and at other times, notation of the use of a statistical test is absent, although reading the text would suggest that a solid conclusion could be made.

In general, the idea that increased Yap levels and activity contributes to overgrowth in the Atg7 KO model is novel and is a substantial contribution to the field. The genetic and pharmacologic data in the various mouse models for a role in Hippo signaling is strong, but the cell biology data does not elucidate a mechanism of how this is accomplished. Further work in this area should be done to make this manuscript acceptable.

I will highlight specific points to be addressed below.

- Olig1-Cre is a novel hepatocyte specific promoter in liver

The authors present several examples of liver cells in a reporter mouse to argue that the Olig1-Cre line is hepatocyte specific. "Representative" single images of various liver cells types is insufficient to fully establish if this Cre line is truly hepatocyte specific. Often, cre lines are not completely penetrant in the intended cell type, nor do they have absolute fidelity. Characterizing the timing of when Olig1 becomes active in the liver is also important for the reader to imagine when the described phenotype may begin.

Preferentially, quantifying the proportion of hepatocytes vs non-parenchymal cell labeling should be performed in the Olig1 cre line. A comparison between hepatocytes and cholangiocytes should also be done, especially since a known deficiency of the Alb-cre line is that there is some early cholangiocyte labeling in that model. This may also be true in the Olig1-cre line. Commonly, quantification is done by manually counting dual or triple labeled cells using IHC or automated counting utilizing FACS from dissociated and labeled tissues (See Yanger et al. (2013), *Gene & Development*. 27:719-24.).

We agree with the reviewer. Since the Olig1-cre line is distracting from the overall main focus of this manuscript we have decided to describe this line in a separate manuscript in the future. We have replicated our data and focused on the use of established liver specific promoters namely Albumin-CRE and ERT2-Albumin-CRE in this current resubmission.

We completely agree with the reviewer; indeed, the Albumin-CRE mice do fate label hepatocytes and cholangiocytes, which is often overlooked in the field. However, the tamoxifen- inducible ERT2-Albumin-CRE fate label only hepatocytes when CRE is induced in adult animals. As the Albumin-CRE/Atg7^{F/F} and the ERT2-Albumin-CRE/Atg7^{F/F} mice had interchangeable phenotypes, we conclude that hepatocytic impairment in autophagy is driving the phenotype and not cholangiocytes.

For the reviewers, we include here immunofluorescence analyses of Albumin-CRE and ERT2-Albumin-CRE mice that had been crossed with Rosa26^{mT/mG} reporter mice (Rosa26-LoxP-Tomato-Stop-LoxP-eGFP) which have ubiquitous tomato fluorescence and upon cre recombination and deletion of the floxed sequence, switch to green fluorescence. While Albumin-CRE/Rosa26^{mT/mG} mice clearly show hepatocyte plus cholangiocytic recombination in adult animals, ERT2-Albumin-CRE/Rosa26^{mT/mG} (injected with tamoxifen at > 8 weeks of age), show complete hepatocytic but not cholangiocytic recombination as early as 7 days after tamoxifen injection.

• "Hepatocytes were hypertrophic and hyperplastic...composed of at least 2-3 cells instead of 1 cell (Fig 1d)..."

The resolution of the presented image is difficult to pictorially appreciate this distinction. A reticulin stain of the control and ATG7 knockout should highlight these described differences for the reader. Also, how old is the presented mouse in this example?

We appreciate the comment and have performed cellular membranous staining highlighting the increased cell composition of hepatocyte trabecular plates (Suppl. Fig. 1F). The mouse in this sample is 3 months of age.

- Figure 2d - The ATG7 KO vs Yap signature GSEA of FDR 0.145 is not significant, contrary to the statement by the authors that "GSEA analysis uncovered enrichment of Yap activation gene signatures...". The MST1/2 KO signature is strongly significant (FDR 0.0) and would support that Hippo signaling is inactivated, but there are also non-Hippo pathway substrates that are direct downstream targets of the MST kinases. It is possible (although unlikely) that non-Hippo pathway targets of MST cause the GSEA to align perfectly with the ATG7 KO. The Yap signature assayed in this figure is derived from breast cancer cells and could explain why the FDR as compared to liver cells is not significant.

Since the authors argue that Yap overexpression is a major reason why the liver is enlarged in the ATG7 knockout model, a GSEA compared against available liver specific Yap overexpression models such as Dong et al. (2007) or Yimlamai et al. (2014) should be presented and would be more relevant.

Unfortunately, gene signatures from Dong et al (2007) and Yimalai et al (2014) are not available.

We were thus not able to perform the requested analyses. However, we believe that we have more data showing robust linkage of autophagy and the Hippo tumor suppressor pathway.

- Figure 2e - "Afp, BIRC5, Gli2, ITGB2" - Have not been shown to be a direct targets of Yap/TEAD by DNA binding assay. These genes have repeatedly been positively associated with Yap activity, but cannot be strictly referred at Hippo target genes. Please provide direct references if this is not correct. Alternatively, other genes such as Jag1 and Notch2 have been shown to be Yap/TEAD targets in the liver and the authors could consider assaying these genes.

We thank the reviewers for their comments. We have removed Afp and Gli2 and have included other target genes: Cyr61 and Itgb2 have been identified as direct targets of Yap by ChIP-Seq⁶. Areg has been identified by ChIP⁷. Birc5 is an accepted bona fide Yap/Taz downstream target^{8,9}. Further Yap target genes identified by Zanconato et al (e.g. Axl, Ctnnb1) and Ctgf (Yimlamai et al 2014) have been included in the array analysis.

- Figure 2f - Please clearly identify the Hippo target genes that are upregulated during ATG7 knockout. As stated above, some genes assayed (ie. Birc5, Afp, Itgb2...) are not strictly Hippo target genes as stated in the text.

Please see above.

- "primary hepatocytes isolated from Olig1-Cre:Atg7f/f mice..." - Details of this procedure could not be found in the methods and would be important to future investigators should they like to repeat these experiments.

We have included details about the procedure in the methods section.

Figure 3d - This set of blots is not visually consistent with the author's statements that Yap and Yap activity (as evidenced by pYap/Yap) is increased in the Olig1-Cre/ATG7 KO model. There

is no loading control in the first set of blots for comparison. Is the reader supposed to refer to the b-tubulin loading control in the second set of blots for reference? If so, then why does the Taz lane appear so significantly different between these two sets of blots.

We have analyzed whole liver lysates from ERT2-Albumin-CRE/Atg7 and found increased Yap protein levels at d7, d14 and d28 after tamoxifen injection. We have provided loading controls (beta tubulin) from corresponding blots (Fig. 1J).

To this reviewer, the Yap levels in controls versus Olig1-cre/ATG7 line appear very similar. The quantification in Figure 3c of Yap activity (p-Yap/Yap) does not show significant difference which does not support the idea of increased Yap activity in this model. In comparison, Taz appears to be markedly enriched in this set of blots as well as in the second presented set. The authors have suggested that Taz does not operate in hepatocytes by their IHC (Fig 3a), so we are left wondering if there is reduced Hippo signaling in the hepatocyte compartment as stated in the text. Please provide darker exposures of P-Mst1/2 and Mob blots.

As Yap has been shown to be sequestered in the cytoplasm (e.g. by dystrophin independent of phosphorylation), we instead focused on the biologic outcome (overall Yap protein levels and genes associated with Yap activation) rather than characterizing the multitude of Yap upstream regulators.

- Supplementary Figure 3e/f - No statistical tests are presented to help the reader evaluate the impact of Cyr61/Afp differences, so the statement that "Cyr61 and Afp demonstrated markedly increased expression in Olig1-cre...Yap activation is a specific effect of autophagy impairment" cannot be seriously considered. The data from this figure suggest that BDL and CCl4 injury also activate Yap activity, in some cases to a similar degree as the Olig1-cre Atg7 knockout. Also, other authors have shown some data to that in the presented models (ie. PHx, Wang C et al 2012./Wu H et al 2013.) Yap activation is occurring. It may be a truer statement that Atg7 knockout more potently activates Yap activity.

We agree. Yap activation is likely a regenerative response in the liver injuries that we analyzed. However, none of the liver injuries (cholestatic liver injury by bile duct ligation (BDL), partial hepatectomy, chronic liver injury by CCl4) induces hepatomegaly that is remotely similar to the massive hepatomegaly observed in Atg7 KO mice (up to 11-fold increase). We thus agree with the reviewers, that Yap activation more potently and sustainedly activates Yap activity. For clarity of the manuscript, we have omitted these data from the current submission.

- "Loss of Atg7 drives Yap-mediated cell proliferation..." - This section of the paper likely is one of the most novel and important portions of the manuscript, but is not well assembled. As a reader, I am not certain how Yap levels/activity increase, only that it does and would be consistent with the general premise of the paper. The general impression that I have, is that dysregulation in autophagy leads to impairment in proteasomal degradation of Yap. This hypothesis, nor an alternative hypothesis of how Yap accumulates in this model is not clearly stated.

We analyzed proteasomal activity in shAtg7 and scram-AML infected cells and found comparable or even increased levels of proteasomal activity (Suppl. Fig. 3D). Thus, we conclude that decreased proteasomal activity is not the underlying cause for increased Yap levels / activity.

"...proteasome, but other inactivating mechanisms have been described." - What are these

"other inactivating mechanisms"? In a literature search I only identified various means of ubiquitination that lead to Yap degradation.

Inactivation of Yap may result via the following mechanisms

1. Degradation
 - a. Proteasomal degradation via phosphorylation, ubiquitination and subsequent degradation ¹⁰
 - b. And degradation via autophagy, which we propose.
2. Inactivation via
 - a. Cytoplasmic sequestration
 - i. Phosphorylation dependent ¹⁰
 - ii. Phosphorylation independent (e.g. via AMOT) ^{11,12}

"...which revealed a significant increase in Yap protein stability (Fig 4d, e)..." - The Yap blots presented for controls and shAtg7 do not appear significantly different. The quantification of these Yap blots in 4e appear different, but there is no indication that there is a statistically significant difference (4h-8h).

We have included the statistical analysis.

"...proteasomal activity was not impaired...it was increased in shAtg7-AML12 cells...similar to features of primary hepatocytes..." - Sup Fig 5 shows a significant increase in proteasomal activity in shATG7 cells, but no difference or even a decrease in such activity in primary hepatocytes. This statement is not consistent with the presented data.

We politely disagree. We think that our findings are consistent with the presented data since they suggest that a degradation pathway other than the UPS system (ie., autophagy) is responsible for stabilization of Yap.

References

1. Czaja, M. J. Function of Autophagy in Nonalcoholic Fatty Liver Disease. *Dig Dis Sci* **61**, 1304-1313 (2016).
2. Mittal, S. et al. Hepatocellular Carcinoma in the Absence of Cirrhosis in United States Veterans is Associated With Nonalcoholic Fatty Liver Disease. *Clin Gastroenterol Hepatol* **14**, 124-31.e1 (2016).
3. Lade, A., Noon, L. A. & Friedman, S. L. Contributions of metabolic dysregulation and inflammation to nonalcoholic steatohepatitis, hepatic fibrosis, and cancer. *Curr Opin Oncol* **26**, 100-107 (2014).
4. Yewdell, J. W., Lacsina, J. R., Rechsteiner, M. C. & Nicchitta, C. V. Out with the old, in with the new? Comparing methods for measuring protein degradation. *Cell Biol Int* **35**, 457-462 (2011).
5. Jansens, A. & Braakman, I. Pulse-chase labeling techniques for the analysis of protein maturation and degradation. *Methods Mol Biol* **232**, 133-145 (2003).
6. Zanconato, F. et al. Genome-wide association between YAP/TAZ/TEAD and AP-1 at enhancers drives oncogenic growth. *Nat Cell Biol* **17**, 1218-1227 (2015).
7. Zhang, J. et al. YAP-dependent induction of amphiregulin identifies a non-cell-autonomous component of the Hippo pathway. *Nat Cell Biol* **11**, 1444-1450 (2009).
8. Dong, J. et al. Elucidation of a universal size-control mechanism in *Drosophila* and

- mammals. *Cell* **130**, 1120-1133 (2007).
9. Shen, Z. & Stanger, B. Z. YAP regulates S-phase entry in endothelial cells. *PLoS One* **10**, e0117522 (2015).
 10. Zhao, B., Tumaneng, K. & Guan, K. L. The Hippo pathway in organ size control, tissue regeneration and stem cell self-renewal. *Nat Cell Biol* **13**, 877-883 (2011).
 11. Moleirinho, S. et al. Regulation of localization and function of the transcriptional co-activator YAP by angiomin. *Elife* **6**, (2017).
 12. Zhao, B. et al. Angiomin is a novel Hippo pathway component that inhibits YAP oncoprotein. *Genes Dev* **25**, 51-63 (2011).

Reviewers' comments:

Reviewer #1 (Remarks to the Author):

The authors have added a significant amount of data and addressed most of my comments, in particular about hepatocarcinogenesis. I have only one major comment:

1. How do the authors explain the increase of YAP during the pulse-chase experiment in Fig.2E. This seems counterintuitive unless the authors CHX affects degradation mechanisms first before affecting translation of YAP. The authors should also check if a sufficiently high dose of CHX was used - as some cells have low sensitivity. It appears that there is almost no decrease of YAP levels during the entire observation period. In Figure 2F, the increase in YAP levels in leupeptin treated cells is not convincing. This should be analysed in multiple replicated and clearly shown. Overall, the data on half-life and YAP degradation by ATG7 remain a weakness.

All other comments are minor and should be easily addressed:

1. The introduction could be shortened.
2. The quality of the YAP IHC in Fig.1E (which is not an easy staining) is not perfect and Fig.1J lacks a quantification of YAP and demonstration that there is a significant increase. The authors might also want to determine expression of its paralog, TAZ, is also increased.
3. Why is there more YAP in the shATG7 cells (which makes sense in view of the proposed autophagy-mediated degradation) in Fig.2F but not in Fig.2B and Fig.2D?
4. Fig.2H is based on a single cell - this should be shown in multiple cells and statistic should be included. Likewise, Fig.6C is based on two cells and relevance is questionable unless the authors show this to occur more commonly.
5. Although differences in hepatocyte size seem quite apparent, the authors should still measure hepatocyte for Fig.3G and show statistics.
6. Does YAP deletion in the Atg7ko mice completely suppress the expression of YAP target genes besides Cyr61? This point is relevant as there can be compensation by TAZ in some settings and deserves to determine more than a single YAP target gene
7. In Fig.5, the in vivo and in vitro experiments should be grouped.
8. There are some typos (e.g. "enewal").

Reviewer #2 (Remarks to the Author):

I have reviewed the revised manuscript and rebuttal letter. The revised manuscript is improved in many respects. The most exciting new results are the human data provided in Figure 6, which provides a more robust correlation between loss of ATG7 and the activation of a YAP transcriptional signature.

My main concern that persists is the author's response to my previous points 2 and 3 with the new data provided in Figure 2. The effects of the pharmacological inhibition studies are on the whole modest and the agents used are non-specific. I don't find these results particularly compelling. Genetic analysis of multiple ATGs is the current standard in the field to establish a general role for the autophagy pathway in mediating a biological phenotype. Importantly, these experiments in

Figure 2 are being conducted using cell lines, not in vivo models, and reagents to knockdown ATGs are readily attainable. Thus, the previously requested genetic loss-of-function approaches against additional ATGs are achievable and such experiments are important to validate that YAP is being degraded via autophagy and to corroborate that YAP activation results from a general deficiency in the autophagy pathway.

Reviewer #3 (Remarks to the Author):

Lee et al. describe the regulatory role that autophagy plays in degrading the oncoprotein Yap in the liver. As a result of autophagy defects, Yap accumulates, resulting in hyperproliferation and oncogenesis. The authors present data regarding p62/Nrf1 as a parallel pathway involved in autophagy that is separate from the accumulation of Yap. This is important as p62 has been implicated as a potent component of the autophagy pathway that results in HCC. Autophagy of YAP appears to be a separate mechanism that contributes to NASH and HCC.

The paper is much clearer than its previous version, focusing on the trafficking of YAP through the autophagy pathway and the consequence if this pathway is impaired. As of yet, there does not appear to be any papers in the literature which support a mechanistic role for controlling Yap levels in this way, making this an important contribution to the literature.

There are some modest changes which are suggested below, which should help in the clarity of the paper, noted below.

-The contrast in Figure 1D is not particularly striking w/o significant enlargement. Contrast for both control and Atg7 KO mouse should be increased. Also, what is highlighted in the box/inset? A normal bile duct? Please clarify.

- Figure S11 - Primary hepatocyte staining for yap generally appears more impressive than the tissue IHC. It would be good to have perform IB or Hippo target gene analysis of the primary hepatocytes in the Atg7 KO.

- "Analysis of whole liver RNA by gene..." - The comparison is unclear and does not specify a subject, please rewrite.

- Fig 2A - In the AML12 cells, there seems to be an increased overall amount of YAP, but its difficult to say there is more nuclear localization from the picture. Also the nuclear/cytoplasmic fractions in Fig 2B do not strongly support their conclusions as Yap seems to be increased in the same ratio as their loading control, Nucleoporin.

I think it would be good practice to show some data from a 2nd hairpin RNA against ATG7 to ensure that what is seen is not due to an off-target effect. Maybe at least a 2nd luciferase assay.

- "...both p62/Sqstm1 and Yap as well as normal..." - Yap expression is lost in Atg7/Nrf2 mice, so this statement that there is a normal protein level is false. What is the effect on Yap of Nrf2 knockout alone? the data would appear to be available as Figure 4C has this data. Are Nrf2 KO generated in a similar way? This also be included in the schematic on Figure 4A and presented in the IB in 4B.

- Discussion - Would appreciate more discussion regarding the HCC subtypes and how YAP/Atg7 contribute to some subtypes and not to others.

- Figure 6C (Legend) - Please describe what the squares in the figures refer to.

Reviewers' comments:

Reviewer #1 (Remarks to the Author):

The authors have added a significant amount of data and addressed most of my comments, in particular about hepatocarcinogenesis. I have only one major comment:

1. How do the authors explain the increase of YAP during the pulse-chase experiment in Fig.2E. This seems counterintuitive unless the authors CHX affects degradation mechanisms first before affecting translation of YAP. The authors should also check if a sufficiently high dose of CHX was used - as some cells have low sensitivity. It appears that there is almost no decrease of YAP levels during the entire observation period. In Figure 2F, the increase in YAP levels in leupeptin treated cells is not convincing. This should be analyzed in multiple replicated and clearly shown. Overall, the data on half-life and YAP degradation by ATG7 remain a weakness.

We agree with the reviewer, in that we observe a relative although not significant increase in Yap levels upon CHX treatment in Fig. 2E. We tested higher doses of CHX (70-100 µg/ml), which impaired cell viability and were thus not informative. While we acknowledge the weaknesses of this assay, we would like to point the reviewers attention to the significant difference in YAP levels upon CHX treatment in shATG7- and scram-cells.

To confirm the effects of autophagy on YAP levels, we have now generated shATG5-cells with knock-down of ATG5, another essential autophagy protein. New data included in this resubmission demonstrate increased YAP protein levels by immunoblotting and increased Tead4-Luciferase activity in shATG5 cells (see Supplemental Figure 4J, K) consistent in range and response to our data in shATG7 cells. While we agree that the increase in YAP levels are modest overall (2-3 fold), this range of increase is entirely consistent with prior studies, for example in samples from patients with hepatocellular carcinoma (Xu et al, 2009 DOI: 10.1002/cncr.24495).

Similarly, the increases in YAP levels in Leupeptin-treated cells are modest but consistent in their magnitude of change with the CHX data.

All other comments are minor and should be easily addressed:

1. The introduction could be shortened.

We have shortened the Introduction as requested.

2. The quality of the YAP IHC in Fig.1E (which is not an easy staining) is not perfect and Fig.1J lacks a quantification of YAP and demonstration that there is a significant increase. The authors might also want to determine expression of its paralog, TAZ, is also increased.

We have included new images of YAP IHC in Fig. 1E, showing both increased cytoplasmic as well as nuclear staining in hepatocytes. We have also included new data of TAZ IHC (Supplemental Fig 3E), which displays increased staining in non-parenchymal cells but little nuclear hepatocyte staining. Immunoblotting of whole liver lysates from mice sacrificed 7 days, 14 days and 21 days after tamoxifen injection (and thus after Atg7 deletion) show an increase of TAZ (Supplemental Fig. 3D), albeit at 14 days as opposed to YAP, which has an earlier increase at 7 days (Fig. 1J).

3. Why is there more YAP in the shATG7 cells (which makes sense in view of the proposed autophagy-mediated degradation) in Fig.2F but not in Fig.2B and Fig.2D?

We agree with the reviewers that the increase in Yap is modest and have thus included densitometry and a representative image showing 2 sets of lysates collected and processed at consecutive passages. Figure 2B shows a consistent increase in nuclear Yap of 1.5- and 2- fold in shAtg7 cells compared to the respective scrambled control.

While we have loaded equal amounts of proteins in all lanes, the increase in Nucleoporin p62 (Nup62) could mislead one to conclude that a higher amount of nuclear protein was loaded in the shATG7 lanes. Increased expression of Nup62 and other nucleoproteins have been described to be cell cycle-regulated, with increases in Nup62 from G1 to G2/M (Chakraborty et al, DOI: [10.1016/j.devcel.2008.08.020](https://doi.org/10.1016/j.devcel.2008.08.020)). This is consistent with our observation that shAtg7 cells have a higher proliferative activity (Fig. 5E) and new Fig. 2 C ($[^3\text{H}]$ -Thymidine incorporation results from 2 independent experiments, measurement in triplicates). The presence of two distinct bands of 61 kD (cytoplasmic precursor of Nup62) and 62 kD (nuclear Nup62) (Davis and Blobel 1986, PMID: 3518946) in the respective fractions further indicate that very little cross-contamination occurred between the cytoplasmic and nuclear fractions.

In Fig. 2D (now Fig. 2E), we agree with the reviewers, but would like to point out that we did not load equal amounts of protein but rather equal amounts of volume, thus accounting for modest experimental variability. Normalized, quantitative analyses clearly show differences between control cells and the experimental cells.

4. Fig.2H is based on a single cell - this should be shown in multiple cells and statistic should be included. Likewise, Fig.6C is based on two cells and relevance is questionable unless the authors show this to occur more commonly.

As requested by the reviewers, we have quantified the co-localization of endogenous YAP and LC3 and have included the data in Fig. 2I, J (cells analyzed n = 9 and 10, respectively). ICC was also quantified in Supplemental Fig. 4F, G (LysoTracker/YAP-DsRed, cells analyzed: n=89 and 57 respectively), to which we assume the reviewers were referring (and not to Fig. 6C, which shows IF analysis of HCC tissue and not immunocytochemistry). We have also included new data analyzing cells transfected with GFP-LC3 and YAP-DsRed, which were incubated with vehicle or Leupeptin/NH₄Cl and found also evidence for colocalization (Supplemental Figure 4H, I, cells n=79 and 69, respectively). Co-localization was analyzed with Image J software and the JACoP plugin.

5. Although differences in hepatocyte size seem quite apparent, the authors should still measure hepatocyte for Fig.3G and show statistics.

We have included new data quantifying hepatocyte size (Fig. 3H and Fig. 3I) which show significant differences between Control and Atg7 KO but not Control and Atg7/Yap DKO.

6. Does YAP deletion in the Atg7ko mice completely suppress the expression of YAP target genes besides Cyr61? This point is relevant as there can be compensation by TAZ in some settings and deserves to determine more than a single YAP target gene.

We have included qPCR analysis of an additional Yap target gene, Areg, in Fig. 4C, to which we assume the reviewer were referring. YAP and TAZ are paralogs but seem to have both overlapping as well as distinct functions. with YAP having a stronger impact on cellular physiology (Plouffe et al., 2018 doi: 10.1074/jbc.RA118.002715). While loss of Yap unequivocally and significantly diminished hepatomegaly (see Fig. 3C) and tumorigenesis, it does not completely abrogate increase in liver size at 4 weeks. We agree that this and also remaining tumorigenesis could be in part be mediated by TAZ and have included this in the discussion.

7. In Fig.5, the in vivo and in vitro experiments should be grouped.

We have changed the sequence of figures (Fig. 5A, B, C, D) which refer now to in vivo data, while Fig. 5 E,F refer now to cell culture data. We have also included “mice” and “cells” in the label of the figures.

8. There are some typos (e.g. “enewal”).

We apologize for the typo and thank the reviewers for pointing this out.

Reviewer #2 (Remarks to the Author):

I have reviewed the revised manuscript and rebuttal letter. The revised manuscript is improved in many respects. The most exciting new results are the human data provided in Figure 6, which provides a more robust correlation between loss of ATG7 and the activation of a YAP transcriptional signature.

My main concern that persists is the author's response to my previous points 2 and 3 with the new data provided in Figure 2. The effects of the pharmacological inhibition studies are on the whole modest and the agents used are non-specific. I don't find these results particularly compelling. Genetic analysis of multiple ATGs is the current standard in the field to establish a general role for the autophagy pathway in mediating a biological phenotype. Importantly, these experiments in Figure 2 are being conducted using cell lines, not in vivo models, and reagents to knockdown ATGs are readily attainable. Thus, the previously requested genetic loss-of-function approaches against additional ATGs are achievable and such experiments are important to validate that YAP is being degraded via autophagy and to corroborate that YAP activation results from a general deficiency in the autophagy pathway.

We thank the reviewer for his/her comments. We have generated shATG5-cells and found increased YAP protein levels and Tead-4-Luciferase activity levels, consistent with YAP activation in another cellular model of autophagy impairment (Supplemental Fig. 4J, K).

Reviewer #3 (Remarks to the Author):

Lee et al. describe the regulatory role that autophagy plays in degrading the oncoprotein Yap in the liver. As a result of autophagy defects, Yap accumulates, resulting in hyperproliferation and oncogenesis. The authors present data regarding p62/Nrf1 as a parallel pathway involved in autophagy that is separate from the accumulation of Yap. This is important as p62 has been implicated as a potent component of the autophagy pathway that results in HCC. Autophagy of

YAP appears to be a separate mechanism that contributes to NASH and HCC.

The paper is much clearer than its previous version, focusing on the trafficking of YAP through the autophagy pathway and the consequence if this pathway is impaired. As of yet, there does not appear to be any papers in the literature which support a mechanistic role for controlling Yap levels in this way, making this an important contribution to the literature.

There are some modest changes which are suggested below, which should help in the clarity of the paper, noted below.

1. The contrast in Figure 1D is not particularly striking w/o significant enlargement. Contrast for both control and Atg7 KO mouse should be increased. Also, what is highlighted in the box/inset? A normal bile duct? Please clarify.

We apologize, due to file size constraints and pdf conversions, the quality of the IF pictures was diminished. The box insets show a magnification of ductular cells.

[redacted]

- "Analysis of whole liver RNA by gene..." - The comparison is unclear and does not specify a subject, please rewrite.

We have rewritten the paragraph to “Gene set enrichment analysis of the liver’s transcriptome”... Please let us know if you would rather suggest a different wording.

- Fig 2A - In the AML12 cells, there seems to be an increased overall amount of YAP, but its difficult to say there is more nuclear localization from the picture. Also, the nuclear/cytoplasmic fractions in Fig 2B do not strongly support their conclusions as Yap seems to be increased in the same ratio as their loading control, Nucleoporin.

We agree with the reviewer and refer to our reply to point 3 of Reviewer 1, who raised similar concerns.

I think it would be good practice to show some data from a 2nd hairpin RNA against ATG7 to ensure that what is seen is not due to an off-target effect. Maybe at least a 2nd luciferase assay.

- We ordered additional shRNA for ATG7 which did not show effective in knock-down. CrisprCas KO clones from two different cell lines unfortunately yielded only partial knock-down. However, we were able to successfully generate shATG5 cells which showed increased YAP levels by immunoblotting and YAP activity by Tead4-Luciferase assay.

- We used two different Tead4-luciferase reporter (8xGTIIC-luciferase, Addgene #34615, published by Dupont, Piccolo et al, 2011, Nature, doi: 10.1038/nature10137) and another luciferase reporter from Dr. Fernando Camargo (kind gift, published in Yimlamai et al 2014, CELL, doi: 10.1016/j.cell.2014.03.060). While fold change increase in control and shAtg7-cells was comparable, the sensitivity of the reporter by Dr. Camargo was about 100-500 fold greater than the one from Dr. Piccolo. Thus, the Tead4-Luciferase assays in the manuscript were performed with the reporter from Dr. Camargo.

- For the reviewers, we have included both a luciferase assay of scram- and shAtg7 cells utilizing the reporter from Dr. Piccolo (left figure below) and also a head-to-head comparison of the two reporters (right figure). To analyze sensitivity, cells had been also transfected with constitutionally active Yap mutants YapS127A and YapS5A.

- "...both p62/Sqstm1 and Yap as well as normal..." - Yap expression is lost in Atg7/Nrf2 mice, so this statement that there is a normal protein level is false. the data would appear to be available as Figure 4C has this data. Are Nrf2 KO generated in a similar way?

We have repeated immunoblotting in Fig. 4B with new tissue samples from Nrf2 KO and Nrf2/Atg7 DKO from our collaborator. We have now included liver lysates from Nrf2 KO mice. Yap expression in Atg7/Nrf2 DKO livers and Nrf2 KO livers is comparable or decreased compared to the protein levels in control mice. P62/Sqstm1 protein levels are only increased in Atg7 KO and Atg7/Yap DKO. Nrf2 KO mice have a whole body knock out of Nrf2. Nrf2/Atg7 KO were generated by crossing Albumin-CRE/Atg7 floxed mice with Nrf2^{-/-} mice, if this is the question the reviewer was asking?

- Discussion - Would appreciate more discussion regarding the HCC subtypes and how YAP/Atg7 contribute to some subtypes and not to others.

We appreciate the interest and have expanded the Discussion.

- Figure 6C (Legend) - Please describe what the squares in the figures refer to.

The squares in the Figure 6C represent magnification of the inserts. We have adapted the legend to reflect this information.

REVIEWERS' COMMENTS:

Reviewer #1 (Remarks to the Author):

All points are sufficiently addressed.

As a last comment, Fig.2J lacks error bars.

Reviewer #2 (Remarks to the Author):

My previous concerns have been satisfied.

Reviewer #3 (Remarks to the Author):

The revised version of this article by Lee et al. have addressed all of my prior concerns with the manuscript. They have produced additional data that clarifies and strengthens the idea that Yap is an important autophagy substrate that leads to cancer when autophagy is inhibited. This is an important and as of yet, unreported contribution to the literature which will open new areas of investigation. I would highly recommend acceptance of this manuscript.

Reviewers' comments:

Reviewer #1 (Remarks to the Author):

All points are sufficiently addressed.

As a last comment, Fig. 2J lacks error bars.

We are glad that the reviewer accepted our revisions.

Regarding Fig. 2J, we did not include error bars as this graph depicts the ratio of [cells with LC3+/Yap+ dots] / [all imaged cells] in percent. Statistics was calculated by assigning a digital value (1 or 0) to each cell for absence or presence of colocalization as assessed by ImageJ software. Statistics was assessed by students t test.

Reviewer #2 (Remarks to the Author):

My previous concerns have been satisfied.

Thank you.

No comments to be addressed.

Reviewer #3 (Remarks to the Author):

The revised version of this article by Lee et al. have addressed all of my prior concerns with the manuscript. They have produced additional data that clarifies and strengthens the idea that Yap is an important autophagy substrate that leads to cancer when autophagy is inhibited. This is an important and as of yet, unreported contribution to the literature which will open new areas of investigation. I would highly recommend acceptance of this manuscript.

Thank you. We highly appreciate it.

No comments to be addressed.